# FM4NPP: A Scaling Foundation Model for Nuclear and Particle Physics

**David Park**[1][*]**, Shuhang Li**[2][*]**, Yi Huang**[1][*]**, Xihaier Luo**[1]**, Haiwang Yu**[2]**, Yeonju Go**[2]**,
**Christopher Pinkenburg**[2]**, Yuewei Lin**[1]**, Shinjae Yoo**[1]**, Joseph Osborn**[2]**, Jin Huang**[2]**, Yihui Ren**[1][,†]

[1]Artificial Intelligence Department, Brookhaven National Laboratory, United States
[2]Nuclear and Particle Physics Department, Brookhaven National Laboratory, United States
[*] These authors contributed equally. [†] Corresponding author, `yren@bnl.gov`

## Abstract

Large language models have revolutionized artificial intelligence by enabling large, generalizable models trained through self-supervision. This paradigm has inspired the development of scientific foundation models (FMs). However, applying this capability to experimental particle physics is challenging due to the sparse, spatially distributed nature of detector data, which differs dramatically from natural language. This work addresses if an FM for particle physics can scale and generalize across diverse tasks. We introduce a new dataset with more than 10 million particle collision events and a suite of downstream tasks and labeled data for evaluation. We propose a novel self-supervised training method for detector data and demonstrate its neural scalability with models that feature up to 188 million parameters. With frozen weights and task-specific adapters, this FM consistently outperforms baseline models across all downstream tasks. The performance also exhibits robust data-efficient adaptation. Further analysis reveals that the representations extracted by the FM are task-agnostic but can be specialized via a single linear mapping for different downstream tasks.

## 1 Introduction

The emergence of large-scale language and vision models Wang et al. (2023) has marked a paradigm shift from specialized neural architectures tailored to individual tasks toward universal, scalable, and multitasking models. These large models, containing billions of parameters and trained through self-supervised learning on massive unlabeled datasets, can be efficiently adapted to diverse downstream tasks, ranging from language translation and code generation to general reasoning. Recognizing their transformative potential, the scientific community has termed these scalable, general-purpose models as *foundation models* (FMs) Bommasani et al. (2021). Among their underpinning features, FMs can employ self-supervised learning on extensive unlabeled datasets, allowing them to develop generalized representations adaptable to various downstream tasks with minimal additional labeled training. However, scientific data often fundamentally differ from natural language or visual data. Hence, the design and implementation of FMs for scientific fields still faces challenges Li et al. (2024); Pyzer-Knapp et al. (2025a).

This work investigates developing FMs created for experimental nuclear and particle physics (NPP), using data from the Relativistic Heavy Ion Collider (RHIC) and the sPHENIX detector Brookhaven National Laboratory (2025). NPP research uses particle colliders, such as RHIC or the Large Hadron Collider (LHC), to explore subatomic phenomena. Discovery of the Higgs boson exemplified the transformational significance of collider-based NPP Collaboration et al. (2012). In particular, RHIC collides heavy ions and polarized protons, facilitating essential studies of quark-gluon plasma and the structure of protons and nuclei Belmont et al. (2024). Commissioned in 2023 Moskowitz (2023), the sPHENIX detector features advanced tracking and calorimetry and generates extensive and complex data. The complexity of collider data and the breakthrough science it enables have motivated exploration of new data processing tools like FMs that employ self-supervised learning. In particular, the high occupancy of hadronic collisions at RHIC or the LHC is particularly challenging for traditional reconstruction algorithms, motivating exploration into new methods.

**Stage 1:** Pretrain `FM4NPP` on a self-supervised autoregressive forecasting

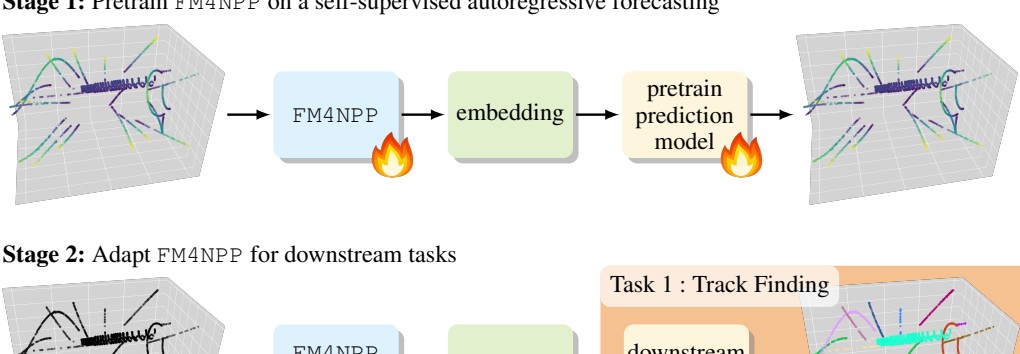

**Stage 2:** Adapt `FM4NPP` for downstream tasks

Figure 1: Overview of a pretrained foundation model that can be adapted to various downstream tasks. We answer two questions in this work: a) Whether the foundation model is *scalable*, i.e., can larger model and dataset sizes improve performance? and b) If the foundation model is *adaptable* to solve multiple downstream tasks.

However, developing an FM for NPP poses several challenges. The sparse, three-dimensional (3D)-spacepoint nature of collider data lacks an established framework for formulating self-supervised tasks. Additionally, optimal neural architectures and the scaling behavior of pretraining losses with respect to model and data size remain unknown. Crucially, it is uncertain if neural representations from a frozen, pretrained FM can generalize effectively to various downstream tasks, thereby outperforming existing traditional solutions and specialized AI models.

Here, we take a first step toward enabling the use of FMs for NPP by adopting a cost-effective two-stage paradigm: 1) pretrain a large FM using a self-supervised objective and 2) pair the frozen FM with lightweight, task-specific adapters (Figure 1). The core hypothesis is a sufficiently trained FM encodes rich, task-agnostic representations that can be efficiently adapted to diverse downstream tasks with minimal additional training.

To this end, we construct a large-scale dataset, exceeding 10 million simulated collision events and characterized by sparse, high-dimensional detector data. We also define three downstream tasks with corresponding labeled datasets to evaluate FM adaptability. We introduce a self-supervised pretraining strategy tailored to the sparsity and structure of detector data and demonstrate strong neural scaling behavior with models up to 188 million parameters. With frozen FMs and simple adapters, we achieve state-of-the-art performance across all downstream tasks. This analysis further reveals that FM representations are broadly task-agnostic and can be specialized using a single linear transformation. In summary, our contributions are:

- A large-scale, open benchmark dataset for FM training and evaluation in particle physics.

- A self-supervised pretraining method designed for sparse detector data.

- Empirical evidence of scaling behavior and data-efficient adaptation with frozen FMs.

- Insight into the structure and adaptability of FM representations across diverse tasks.

## 2 RELATED WORK

**Scientific Foundation Models.** Developing FMs for scientific domains is a promising yet formidable endeavor. Progress has been most evident in domains where data exhibit modality structures similar to language or vision. For instance, Aurora Bodnar et al. (2025) is an atmospheric FM trained on continuous spatiotemporal climate data, and recent work has demonstrated FM-based disease detection from retinal images Zhou et al. (2023). In high energy physics, recently developed FMs, such as OmniJet-$\alpha$ Birk et al. (2024) and OmniLearned Bhimji et al. (2025), focus on high-level physics objects known as jets – collimated sprays of particles that can be represented as dense matrices. However, many scientific disciplines, including low-level detector data (e.g., raw hits or clusters) in particle physics, materials science, and single-cell omics, present unique challenges like irregularly structured and sparse data. Traditional approaches, such as graph neural networks (GNNs), are well suited for sparse data, but they face scalability issues due to certain phenomena, e.g., oversmoothing Rusch et al. (2023). Surveys in materials science Pyzer-Knapp et al. (2025b) and single-cell omics Ma et al. (2024) emphasize additional bottlenecks, including limited data availability and high computational costs. These challenges also apply to NPP, where it remains unclear how best to model extremely sparse data, how much data are needed, and whether pretraining benefits can effectively transfer to downstream tasks. This work takes a first step toward addressing these questions by developing a scalable FM for sparse lower-level detector data in NPP, focused on efficient pretraining, architectural scalability, and downstream generalization.

**Scalable Neural Architectures.** Three neural network architectures are prominent for their scalability: Transformers, Mixture-of-Experts (MoE), and State Space Models (SSMs). The Transformer architecture Vaswani et al. (2017) has revolutionized deep learning via self-attention, enabling effective modeling of long-range dependencies. This has led to widespread adoption in both natural language processing (NLP) and computer vision Dosovitskiy et al. (2021). However, the quadratic time and space complexity of self-attention limits scalability on long sequences – a critical bottleneck for scientific data. MoE architectures Fedus et al. (2022) improve inference efficiency by activating only a subset of the model per input, although they face challenges, such as training instability and expert imbalance. The Mamba architecture Gu & Dao (2024), an SSM variant, achieves linear time complexity and shows competitive or superior performance to Transformers. Given the relatively large number of spacepoints per collision event, which can result in especially long sequences, we explore SSMs as a backbone due to their favorable training efficiency and memory usage.

**AI Model Tasks in NPP.** In collider physics, high-energy particles collide to produce new particles, whose trajectories, called *tracks*, are reconstructed from discrete spacepoints recorded by layered detector components. Track finding, or assigning spacepoints to different tracks, is one of the most important tasks. Traditional algorithms rely on combinatorial seeding followed by Kalman-filter-based refinement Kalman (1960). These classical methods are computationally expensive and difficult to parallelize on modern accelerators. GNN-based approaches have become popular in track finding. `Exa.TrkX` Ju et al. (2021) formulates the task as edge classification, whereas `EggNet` Calafiura et al. (2024) employs contrastive learning followed by clustering. Each predicted track then corresponds to a connected subgraph of spacepoints. Other recent work has introduced Transformer-based models, such as `HEPT` Miao et al. (2024), and SSM-based Jiang & Qian (2025) tracking models at $\mathcal{O}(1M)$ parameter scale. Beyond track finding, another common task is particle identification (PID). Previous machine learning (ML)-based PID approaches, such as MLPF Mokhtar et al. (2025) and HGPF Kakati et al. (2025), rely on reconstructed tracks as key inputs and typically incorporate calorimeter topoclusters. In contrast, we employ a lightweight adapter model, designed as a probe of FM generalizability, that performs PID directly at the low level using only time projection chamber (TPC) spacepoint data without requiring high-level track information or calorimeter inputs.

While these models are promising – with some achieving robust results using fewer than one million parameters – there is no systematic study of scaling behavior. Moreover, open datasets designed for scaling and evaluating FMs are limited. In this work, more than 10 million simulated collision events are generated, affording comprehensive scaling studies. We also develop an FM with 188 million parameters, two orders of magnitude larger than prior models, and evaluate its performance in tracking and for other broader NPP tasks.

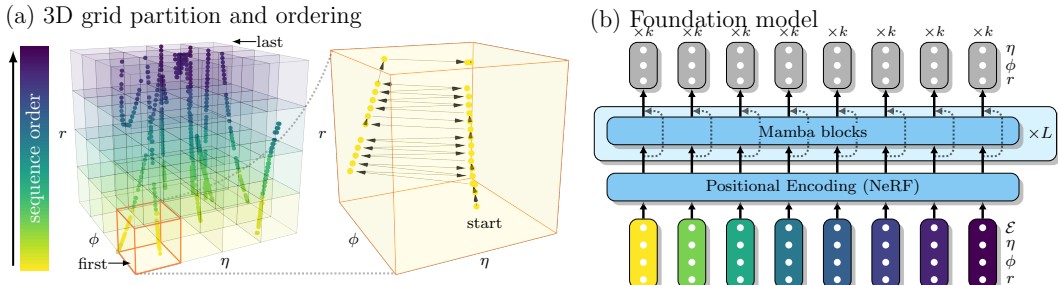

Figure 2: (a) Hierarchical Raster Scan strategy to serialize the unordered spacepoints into a one-dimensional sequence. (b) A Mamba FM backbone for k-Next-Nearest-Neighbor prediction.

## 3 PARTICLE DETECTOR DATASET

**Dataset.** As part of the sPHENIX detector's central tracking system, the high-granularity TPC Klest (2020) records more than $85\%$ of the total data volume. The TPC consists of 48 concentric cylindrical readout layers, encompassing approximately 160,000 channels that each record 260 time samples, totaling 41.6 million voxels. Functioning as a three-dimensional (3D) camera, the TPC records the paths of particles emerging from collision events, delivering continuous 3D spacepoint information.

We use a publicly available dataset with over 10 million simulated p+p collision events at $\sqrt{s} = 200$ GeV Li et al. (2025). The natural sparsity of events in p+p collisions makes them an ideal testing ground for developing an FM for NPP applications. The simulation pipeline includes real detector geometry, electromagnetic fields, hadronic interactions, continuous energy loss, multiple scattering, decay processes, secondary particle production, and precise energy deposition. The raw detector hits subsequently are reconstructed to spacepoints that serve as inputs in this work. More concretely, a collision **event** $E$ is represented as a set of **spacepoints** $\{s_i\}$, where each spacepoint is expressed by its deposit energy and location $(\mathcal{E}, x, y, z)$. The number of spacepoints per event can vary from hundreds to thousands.

**Downstream Tasks.** We select three complementary downstream tasks to evaluate the generalizability of an FM: **Track Finding**, **PID**, and **Noise Tagging**. **Track Finding** assigns each spacepoint to its corresponding predicted track as shown in Figure 1 (Task 1, 2, and 3). Assume there are $m$ **tracks** $\{T_j\}_{j=1}^m$, where each track $T_j$ consists of its associated spacepoints $\{s_i \in T_j\}$. The goal of track finding is to predict a partition $P$ over the set of spacepoints, where $P_i^j = 1$ if spacepoint $s_i$ is assigned to track $T_j$. The number of tracks can vary from event to event. This task is analogous to instance segmentation in computer vision.

To evaluate performance, we employ both conventional physics-motivated metrics, **tracking efficiency and purity** Calafiura et al. (2018), as well as the statistical metric Adjusted Rand Index (ARI) Hubert & Arabie (1985). As the exact definition of whether or not a predicted track matching a true track can differ among physics experiments, we adopt the "double-majority rule" from the TrackML challenge Amrouche et al. (2020). The rule enforces that a predicted track is successfully matched to a true track only when greater than 50% of the predicted track's spacepoints belong to that track and more than 50% of the true track's spacepoints are present in the predicted track. This stringent rule guarantees neither predicted tracks nor true tracks are matched more than once. Then, tracking efficiency (recall) is defined as the ratio between the true positive and total number of truth tracks, while tracking purity (precision) is the ratio between the true positive and total number of predicted tracks.

**PID** aims to label each spacepoint to the particle species that produced it, i.e., pion, kaon, proton, electron, and others. This is comparable to a segmentation task in computer vision. **Noise Tagging**, the third downstream task, seeks to identify spacepoints associated to low-momentum secondary particles, primarily delta electrons as they typically are not associated with physics observables of interest. This also can be considered a segmentation task. For these two downstream tasks, we report overall accuracy, macro-averaged precision, and recall. Additional information about the TPC detector, data generation pipeline, and statistical analysis are included in Appendix A.

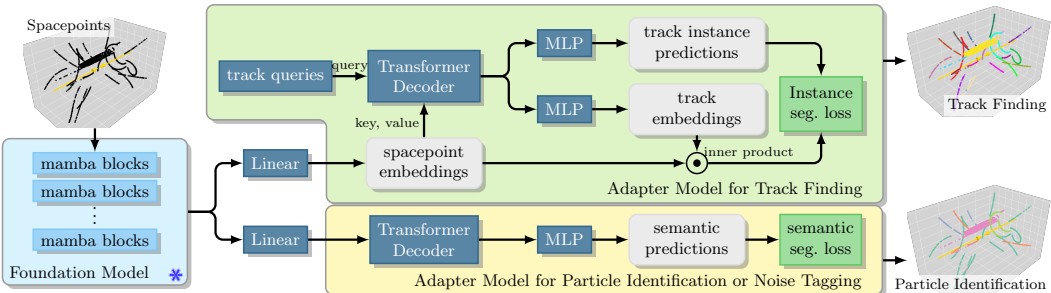

Figure 3: The pretrained FM is kept frozen during training of the adapter models for downstream tasks. The adapter models for particle identification and noise tagging share the same architecture but are trained independently.

## 4 METHODOLOGY

This section introduces the scalable FM for NPP, including a novel serialization method for sparse spacepoints, adaptation to the Mamba architecture, a self-supervised pretraining objective, and lightweight adapter models for downstream tasks. Additional information is included in Appendix B.

### 4.1 SELF-SUPERVISED SCALING FOUNDATION MODEL

**Serialization of Spacepoints.** A key challenge in applying sequence-based models like Mamba to particle detector data is in serializing the unordered set of 3D spacepoints $s_i$ from an event $E$ into a meaningful one-dimensional (1D) sequence. The serialization strategy must balance two competing objectives: preserving the *global* structure of particle trajectories, which typically propagate outward from the collision point, and maintaining *local* continuity along individual tracks $T_j$ to retain fine-grained geometric information. Naive serialization schemes struggle to achieve this balance. For example, space-filling curves (e.g., Hilbert or Z-order) prioritize spatial locality but often interleave points from different tracks, disrupting trajectory coherence. Conversely, sorting points by their radial distance preserves the outward particle flow but scatters spacepoints from the same track across distant positions in the sequence, breaking local continuity. An effective serialization must navigate this trade-off, allowing the model to learn both *global* and *local* physics from a sequential input.

We propose a *Hierarchical Raster Scan* strategy to serialize the unordered spacepoints $s_i$ into a 1D sequence suitable for sequence models as shown in Figure 2(a). First, all spacepoints are transformed from Cartesian $(x, y, z)$ to a cylindrical-polar system $(r, \phi, \eta)$ that better reflects the geometry and symmetries of collider experiments, where $r$ is radial distance, $\phi$ depicts the azimuthal angle, and $\eta$ represents the pseudorapidity (angle to the beam axis). The raster scan method operates on two levels. The first is *inter-box ordering*, where spacepoints are initially partitioned into non-overlapping 3D spatial boxes. Then, these boxes are ordered based on the $(r, \phi, \eta)$ coordinates of their geometric centers, starting from the innermost region and progressing outward. This produces a global ordering over the spatial domain. The second is *intra-box ordering*, where, within each box, spacepoints are sorted by their radial coordinate $r$, which generally aligns with the direction of particle propagation. By concatenating the intra-box sequences according to the inter-box order, we obtain a globally serialized sequence that preserves both *local* spatial continuity and *global* physical progression. This hierarchical structure captures important geometric and physical priors while producing a format compatible with sequence models. Specifically, we partition the spatial domain into a $6 \times 8 \times 8$ grid along the $(r, \eta, \phi)$ axes, respectively. The $r$ bins are aligned with the physical boundaries of the TPC detector layers, while the $\eta$ and $\phi$ bins are determined using frequency-based binning to ensure balanced point distributions across the grid.

**Mamba as the FM Model Backbone.** Mamba is a selective SSM that efficiently processes long sequences, achieving linear time complexity Gu & Dao (2023). It features a selection mechanism that makes its internal state matrices input-dependent, allowing the model to dynamically focus on relevant information and filter out noise – all while using a hardware-aware algorithm for fast computation. In this work, we employ Mamba-2 Dao & Gu (2024), which further improves upon this foundation. Mamba-2 introduces structured State Space Duality (SSD), a new theoretical framework

that simplifies the architecture and enhances hardware utilization, leading to significant speedups in both training and inference.

We treat every spacepoint as an input "token" in a sequence. To map an input tuple $(\mathcal{E}, r, \phi, \eta)$ to the model width $d_{\mathrm{model}}$, we employ a two-pathway process inspired by Neural Radiance Fields Mildenhall et al. (2021). First, the feature component $\mathcal{E}$ is projected into a feature embedding of dimension $d_{\mathrm{model}}$. Concurrently, the spatial coordinates $(r, \phi, \eta)$ are transformed with a high-frequency positional encoding function, $\gamma(\cdot)$, which uses sine and cosine transformations. Then, this encoded position is also projected into a positional embedding of dimension $d_{\mathrm{model}}$. The final representation is the element-wise sum of the feature and positional embeddings, yielding a single vector of size $d_{\mathrm{model}}$ that holistically captures the event's properties and location.

**Self-supervised Pretraining Objectives.** To create a self-supervised pretraining task, the prediction objective must be decoupled from the sequence order as a naive "next-spacepoint prediction" would learn artifacts of the serialization itself. The target for any given spacepoint $s_i$ must be defined by its geometric relationship to other spacepoints in 3D, not its 1D sequence position. While predicting nearest neighbors is a natural geometric objective, a standard k-Nearest-Neighbor task is unsuitable in an autoregressive framework due to information leakage from previously seen spacepoints. We partially address this by introducing *k-Next-Nearest-Neighbor prediction* (k-NNN) (Figure 2(b)), which aligns the objective with particle propagation. For any query spacepoint $s_i$, the model's task is to predict its $k$ nearest spacepoints within its next neighborhood $\mathcal{N}_c(s_i) = \{s_j \in E \mid r_j > r_i\}$, i.e., those with larger radius. Let $\widehat{\mathbf{Y}}_i = \{\widehat{\mathbf{y}}_{i,1}, \ldots, \widehat{\mathbf{y}}_{i,k}\}$ denote the predictions and $\mathbf{Y}_i = \{\mathbf{y}_{i,1}, \ldots, \mathbf{y}_{i,k}\}$ the ground truth neighbors, both ordered by increasing distance. The loss is then $\mathcal{L}_i = \frac{1}{k} \sum_{m=1}^{k} \|\widehat{\mathbf{y}}_{i,m} - \mathbf{y}_{i,m}\|_2^2$. Larger $k$ expands the geometric horizon and makes the task more difficult.

## 4.2 Adaptive Models for Downstream Tasks

**Track Finding.** Figure 3 depicts how our downstream adapter model for track finding, formulated as an instance segmentation task, is inspired by image panoptic segmentation models Cheng et al. (2021; 2022) and adapted to sparse spacepoints data.

Point-level features from the FM are first projected to spacepoint embeddings via a single linear layer. This projection serves as a task-alignment filter, compressing and reorienting the pretrained representation into a lower-dimensional space while also providing a probing point to assess the task relevance of the FM features. We initialize $N$ learnable queries (track queries) $\mathbf{Q} = \{\mathbf{q}_k\}_{n=1}^{N}$ and refine them over $L$ transformer decoder layers. In each layer, cross-attention aggregates information from spacepoint embeddings, modulated by an additive attention mask computed from intermediate assignment logits and followed by self-attention among the queries. The resulting refined track queries are passed through two separate multilayer perceptron (MLP) heads to produce a track embedding and a classification score $\hat{y}_n$. Point-to-query assignment probability $\hat{p}_{in}$ is computed as the sigmoid of the dot product between spacepoint embeddings and each track embedding.

Let $E = \{T_j\}_{j=1}^{M}$ be the set of true tracks of an event $E$. We match the refined track queries to $E$ via the Hungarian algorithm, minimizing the combined cost of Dice loss $\mathcal{L}_{\mathrm{dice}}$, Focal loss $\mathcal{L}_{\mathrm{focal}}$ on the per-point assignments, and classification loss $\mathcal{L}_{\mathrm{cls}}$ for track versus no-object. For each matched pair $(T_j, \mathbf{q}_n)$, the loss is $\mathcal{L}_{\mathrm{match}}^{(j,n)} = \lambda_{\mathrm{dice}} \mathcal{L}_{\mathrm{dice}}^{(j,n)} + \lambda_{\mathrm{focal}} \mathcal{L}_{\mathrm{focal}}^{(j,n)} + \lambda_{\mathrm{cls}} \mathcal{L}_{\mathrm{cls}}^{(n)}$. Unmatched track queries incur only $\mathcal{L}_{\mathrm{cls}}^{(n)}$. We also apply auxiliary losses at each decoder layer. At inference time, each spacepoint $i$ is assigned to the track $n_i^* = \arg\max_n (\hat{p}_{in} \cdot \hat{y}_n)$ and labeled accordingly.

**Particle Identification and Noise Tagging.** For both PID and noise tagging tasks, illustrated in Figure 3, our lightweight adapter first projects each $d$-dimensional point feature into a $d_p$-dimensional embedding via a linear layer then aggregates global context with a single self-attention layer. Finally, it feeds the result through an MLP classifier.

## 5 Experiments and Results

Here, we begin by examining the scaling behavior of our FM with respect to model size, dataset size, and computational cost. We then benchmark the FM paired with lightweight adapters against strong baselines across three downstream tasks. Finally, we present additional analyses to better understand the FM's adaptation behavior.

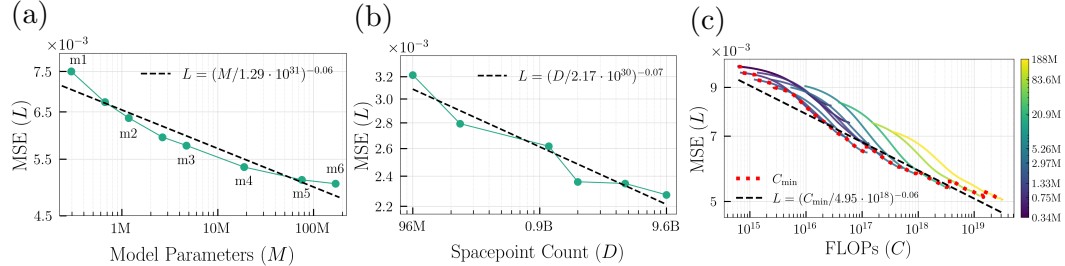

Figure 4: Neural Scaling Behaviors of FM4NPP. We evaluate neural scaling trends on increasing (a) model parameter size $M$, (b) training spacepoint count $D$, and (c) compute in FLOPs. $C_{\min}$ denotes the minimum $L$ for each compute.

## 5.1 NEURAL SCALING BEHAVIORS OF FM4NPP

Table 1: Model Sizes and Compute Resources

| | Model Sizes | | | | | | Compute Resources | | | | | |
|---|---|---|---|---|---|---|---|---|---|---|---|---|
| | m1 | m2 | m3 | m4 | m5 | m6 | NVIDIA GPU | H100 80GB | A100 80GB | | | |
| Model Width | 64 | 128 | 256 | 512 | 1024 | 1536 | Num GPUs | 1 | 1 | 4 | 8 | 24 | 64 |
| Model Params | 0.34M | 1.3M | 5.3M | 21M | 84M | 188M | Train Hrs | 10 | 12 | 20 | 32 | 50 | 72 |

Table 2: Performance on Track Finding. Results for the FM4NPP (m6) model are averaged over 10 random seeds. Uncertainties shown in parentheses indicate standard deviation of the mean in the last digit. The best-performing results for each metric are highlighted in bold, while the second-best results are underlined.

| model | #trnbl para. | Track Finding | | |
|---|---|---|---|---|
| | | ARI↑ | efficiency↑ | purity↑ |
| EggNet | 0.16M | 0.726 | 74.2% | 75.1% |
| Exa.TrkX | 3.86M | 0.877 | 91.8% | 66.4% |
| HEPT | 0.31M | 0.831 | 81.2% | 78.0% |
| AdapterOnly | 2.39M | 0.724 | 78.0% | 64.5% |
| FM4NPP(m6) | 2.39M | **0.945(3)** | **96.1(2)%** | **93.1(1)%** |

Table 3: Performance on Particle Identification and Noise Tagging. Results for the FM4NPP (m6) model are averaged over 10 random seeds. Uncertainties shown in parentheses indicate standard deviation of the mean in the last digit. The best-performing results for each metric are highlighted in bold, while the second-best results are underlined.

| model | #trnbl para. | Particle Identification | | | Noise Tagging | | |
|---|---|---|---|---|---|---|---|
| | | acc.↑ | recall↑ | pre.↑ | acc.↑ | recall↑ | pre.↑ |
| GATConv | 0.91M | 0.692 | 0.397 | 0.637 | 0.910 | 0.673 | 0.806 |
| GCNConv | 0.91M | 0.689 | 0.391 | 0.632 | 0.910 | 0.673 | 0.804 |
| SAGEConv | 0.91M | 0.726 | 0.456 | 0.650 | 0.917 | 0.723 | 0.817 |
| GraphConv | 0.91M | 0.7079 | 0.4176 | 0.6425 | 0.9190 | 0.7213 | 0.8252 |
| OneFormer3D | 44.95M | 0.770 | 0.490 | 0.577 | 0.965 | **0.940** | 0.895 |
| AdapterOnly | 0.74M | 0.663 | 0.339 | 0.611 | 0.911 | 0.622 | 0.836 |
| FM4NPP(m6) | 0.74M | **0.904(1)** | **0.765(3)** | **0.878(3)** | **0.971(1)** | 0.937(1) | **0.919(1)** |

We evaluate our FM's scaling behavior across three axes: model size, dataset size, and compute budget. Results are summarized in Figures 4(a–c).

**Model Scaling.** We construct a series of FMs with varying capacities, denoted `m1` through `m6` in Table 1. Figure 4(a) shows the validation mean squared error (MSE) plotted against model size on a log-log scale, revealing a clear power-law relationship. As the number of parameters increases, validation loss consistently decreases, which aligns with neural scaling laws observed in language and other scientific domains Kaplan et al. (2020); Hoffmann et al. (2022); Nguyen et al. (2023); Bodnar et al. (2025). Notably, performance plateaus at `m6`, suggesting a possible saturation point, which we leave for future investigation.

**Data Scaling.** To isolate the effect of training dataset size, we train the `m3` model on varying subsets (1%, 2.4%, 11.6%, 20%, 47.6%, 100%) of the full dataset. Figure 4(b) shows how performance improves steadily with more data, again following a power-law trend. This suggests the FM can continue to benefit from the large-scale data routinely produced in collider experiments.

**Compute Scaling.** Finally, we study the relationship between compute and model performance. Figure 4(c) shows validation MSE against the total number of floating-point operations (FLOPs) used during training. Models up to `m3` are trained at 25%, 50%, and 100% of their total iteration budget, while larger models (`m4`-`m6`) are trained to full completion only. The results show that smaller models are initially more compute-efficient, but larger models outperform them when more resources are allocated. This highlights the importance of compute-optimal model scaling for deployment in high-throughput environments. All experiments have been conducted using A100 and H100 GPUs with corresponding hardware costs summarized in Table 1.

All models are trained with a batch size of 256. An optimal learning rate of $2 \times 10^{-4}$ is selected through hyperparameter tuning on the `m3` model and reused across all variants using the μ-parameterization principle Vankadara et al. (2024). This approach ensures consistent gradient flow across model sizes and enables zero-shot hyperparameter transfer Yang et al. (2022). Smaller models (`m1`, `m2`) are trained for 50,000 iterations, while larger models (`m3`–`m6`) are trained for 100,000 iterations. We apply cosine learning rate decay with 10,000-step linear warmup and use gradient clipping at a threshold of 0.1. All experiments employ the AdamW optimizer Loshchilov & Hutter (2017) with a weight decay of 0.01. Additional training details are provided in Appendix B.

## 5.2 Performance on Downstream Tasks

**Track Finding.** Our baseline coverage prioritizes reproducible, end-to-end pipelines. Some recent SSM- and Transformer-based trackers cited in "Related Work" are therefore omitted as they either do not report full end-to-end results or lack public, runnable releases. Thus, we focus on `Exa.TrkX` Ju et al. (2021) and `EggNet` Calafiura et al. (2024), both GNN-based methods designed specifically for tracking, as well as `HEPT` Miao et al. (2024), a Transformer-based method that uses locality-sensitive hashing for efficient attention. Because their original implementations target different detector geometries, we adapt them to this dataset (see Appendix C). To verify embeddings extracted using the pretrained FM provide richer information, we also train the lightweight adapter model alone.

Table 2 reports the track finding results of our FM with several baselines. All metrics are computed over the entire test set rather than averaged per event. For example, tracking efficiency is defined as the fraction of all true tracks in the dataset that are successfully matched. Our model achieves higher performance on conventional clustering metrics, such as ARI, and outperforms other approaches in tracking efficiency (recall) and purity (precision).

We also compare this work against the official sPHENIX reconstruction pipeline, which employs a Cellular Automaton seeding followed by a Kalman filter Osborn et al. (2021). As that algorithm is optimized for high transverse momentum ($p_{\mathrm{T}}$), long tracks within the TPC acceptance, we restrict this comparison to tracks that leave at least 20 spacepoints in the TPC and satisfy $p_{\mathrm{T}} > 1\,\mathrm{GeV}$ and $|\eta| < 1.1$. Under these criteria, our model reaches a tracking efficiency of 99.6%, exceeding the sPHENIX pipeline's 94.6%.

**Particle Identification and Noise Tagging.** For the PID and noise tagging tasks, we experiment with four conventional GNN models and report on the best performing one, `SAGEConv`. The graph edge set is constructed by $k$-nearest neighbors with a distance cap. We also adapt and train a state-

of-the-art segmentation model for 3D point cloud data named `OneFormer3D` Kolodiazhnyi et al. (2024).

Table 3 reports the segmentation accuracy, as well as macro-averaged recall and precision. For the PID task, our FM consistently outperforms all baselines, achieving the highest accuracy, recall, and precision. Meanwhile, for the noise tagging, the FM outperforms all GNN-based baselines with similar performance compared to `OneFormer3D`. It is worth noting that `OneFormer3D` has about 45 million trainable parameters, whereas our adapter head has 0.74 million. More details about comparative model implementations and sample outputs are provided in Appendix C.

## 5.3 Insights about FM Adaptation

We aim to further understand FM adaptation behaviors by answering the following questions:

- **Q1**: Does increasing the size of the FM lead to improved performance on downstream tasks?
- **Q2**: Do larger FMs require fewer labeled examples to achieve comparable performance (i.e., better data efficiency)?
- **Q3**: Are the learned FM embeddings task-agnostic, and, if so, how much adaptation is needed to specialize them for specific tasks?

(a) Performance vs. FM model size          (b) Performance vs. dataset size

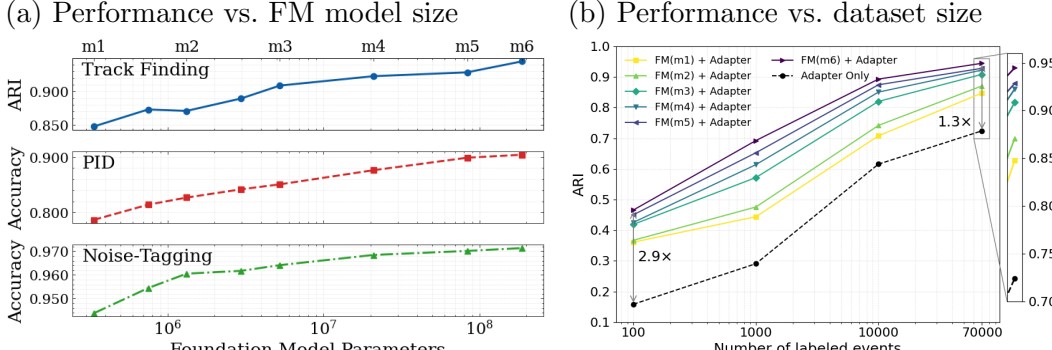

Figure 5: The effect of FM model size (a) and dataset size (b) on downstream task performance.

Figure 5(a) shows the downstream performance of all three tasks plotted against a pretrained FM size. All tasks use the same frozen pretrained representation. Larger pretrained models consistently yield higher performance across every task, confirming that scaling up the pretrained model size improves on various downstream performance even when only the lightweight decoder head is trained.

Figure 5(b) depicts the ARI for the track finding task training on different numbers of labeled data, from 100 to 70,000. Larger FMs consistently outperform smaller ones across different levels of labeled data, indicating neural embeddings extracted from larger FMs contain richer information and can be generalized easier. This confirms common empirical observations that larger models can generalize better Novak et al. (2018). In addition, compared to a baseline adapter-only model trained solely on labeled data (dashed line in Figure 5(b)), pretraining a self-supervised FM model on a large amount of unlabeled data – easy to come by in NPP – proves effective. The relative gain in ARI is greater in the fewer labeled data regions compared to those with abundant labeled data: $2.9\times$ versus $1.3\times$.

## 5.4 Ablation Studies

All ablations use our second-largest model `m5` and identical training budgets. We report absolute changes in task metrics and the relative increase in the remaining gap to perfect performance in Table 4. For bounded metrics such as accuracy and ARI, this normalizes small absolute changes near saturation.

**Neighborhood Size** $k$**.** Our baseline uses $k=30$ during pretraining for the k-NNN objective. Smaller neighborhoods ($k=1$ or $k=5$) reduce downstream performance as the model is conditioned

Table 4: Ablation study results on downstream tasks' performances. Numbers show absolute drops in metric with relative increases in the remaining gap to perfect performance (in parentheses). "Hilbert" represents Hilbert space-filling curve.

| Ablation | Noise Tagging (Acc.) | PID (Acc.) | Track Finding (ARI) |
|---|---|---|---|
| Next-token prediction (vs. k-NNN) | −0.0010 (4.6%) | −0.0023 (2.5%) | −0.0009 (1.6%) |
| k = 1 (vs. k = 30) | −0.0012 (5.7%) | −0.0049 (5.3%) | −0.0019 (3.3%) |
| k = 5 (vs. k = 30) | −0.0007 (3.6%) | −0.0016 (1.7%) | −0.0003 (0.5%) |
| Hilbert (vs. Hierarchical Raster Scan) | −0.0014 (7.0%) | −0.0075 (8.0%) | −0.0051 (9.1%) |

only on very local geometry and misses longer-range structure. In contrast, a moderately larger $k$ allows the FM to capture richer global context while still preserving locality. *Takeaway:* incorporating broader geometric neighborhoods during pretraining produces more transferable representations.

**k-NNN versus Next-token Prediction.** Autoregressive next-token prediction is a widely used strategy in vision and language van den Oord et al. (2016); Radford et al. (2018). Thus, we compare against it to isolate the value of geometry-aware neighborhoods. *Takeaway:* conditioning on local geometric neighborhoods (k-NNN) yields more transferable representations than generic next-token training.

**Serialization Strategy.** Space-filling curves are common for locality-preserving serialization in imaging/point-cloud pipelines Chen et al. (2022). We test the popular Hilbert ordering against Hierarchical Raster Scan (Sec. 4.1) serialization. *Takeaway:* trajectory-consistent serialization beats purely spatial locality – interleaving tracks harms downstream coherence.

## 6 CONCLUSION AND FUTURE WORK

With this work, we demonstrate that FMs can be effectively extended to experimental particle physics by introducing a scalable self-supervised training strategy tailored to sparse detector data. Our model, trained on more than 10 million events, generalizes across diverse downstream tasks with frozen weights and lightweight adapters, consistently outperforming task-specific baselines. Its effective performance and data efficiency suggest the model learns rich, task-agnostic representations that are easily adapted using simple mappings. These findings reveal the potential for general-purpose, scalable models in NPP.

**Limitations and Future Work.** While our work establishes a proof of principle for scaling foundation models on sparse detector data, we acknowledge limitations regarding the scope of our current evaluation. We demonstrate this approach using a single collider experiment (sPHENIX). Transforming this into a universal FM, spanning diverse detector systems, multiple facilities (e.g., LHC), and various collision systems, will require significant future research and community-wide data curation. Additionally, with TPC-only inputs, the number of well-defined downstream tasks is naturally limited. Although we tested generalizability on three downstream tasks, we recognize a broader scope of multi-level and multi-modal tasks with multiple detector system is necessary to fully stress test the model's understanding. Most importantly, validating this approach on real experimental data remains essential to realizing the full potential of this technology for the NPP community.

## ACKNOWLEDGMENTS

The authors would like to express their sincere gratitude to the sPHENIX Collaboration for sharing the simulation data and experimental knowledge, as well as Jubin Choi, Abhay Deshpande, Alexei Klimentov, Michael Begel, Torre Wenaus, Nicholas D'Imperio, James Dunlop, and John Hill from Brookhaven National Laboratory for their valuable support and feedback. This work was supported by the Laboratory Directed Research and Development (LDRD) Program at Brookhaven National Laboratory, LDRD 25-045, which is operated and managed for the U.S. Department of Energy Office of Science (DOE-SC) by Brookhaven Science Associates under contract No. DE-SC0012704. Shuhang Li was partially supported by the DOE-SC through the Office of Nuclear Physics under Award No. DE-FG02-86ER40281. Yihui Ren, Xihaier Luo, and Shinjae Yoo were partially supported by the DOE-SC through the Office of Advanced Scientific Computing Research and the Scientific Discovery through Advanced Computing (SciDAC) program. This research also utilized

resources of the National Energy Research Scientific Computing Center (NERSC), a DOE-SC User Facility, under NERSC Award DDR-ERCAP0034059, and the authors are grateful to the NERSC staff for their support, particularly Shashank Subramanian and Wahid Bhimji.

## REPRODUCIBILITY STATEMENT

We provide anonymized source code and run scripts in the supplementary material to reproduce pretraining and all downstream experiments. Hyperparameters, training schedules, and model sizes (m1–m6) are specified in Sec. 5.1 and Table 1; dataset provenance, preprocessing, and labeling rules are in Appx. A; architectural/serialization details and optimization settings (AdamW, batch size 256, learning rate $2 \times 10^{-4}$ with cosine decay and 10k warmup, gradient clipping 0.1) are in Sec. 4 and Appx. B; evaluation metrics and the TrackML double-majority matching protocol are described in Sec. 3; and baseline adaptations plus additional results are in Appx. C and Tables 2–3. We fix random seeds for all reported runs and include configuration files to replicate numbers. Due to storage limits, we do not include pretrained checkpoints, but the provided scripts reproduce them. Hardware used (A100/H100) is reported in Table 1, and exact environment specs and seeds are listed in the supplementary run scripts.

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

## A    DATASET

The dataset used in this work is based on simulated proton–proton (p+p) collisions at a center-of-mass energy of $\sqrt{s} = 200\,\text{GeV}$, corresponding to conditions of the sPHENIX experiment at the Relativistic Heavy Ion Collider (RHIC). Charged-particle trajectories are recorded with the Time Projection Chamber (TPC). p+p collisions serve as a precision workhorse for testing quantum chromodynamics (QCD) and nucleon structure and provide the baseline for quantifying how particle production in heavy-ion collisions, viewed as a superposition of p+p interactions, is modified by the quark–gluon plasma (QGP) Busza et al. (2018).

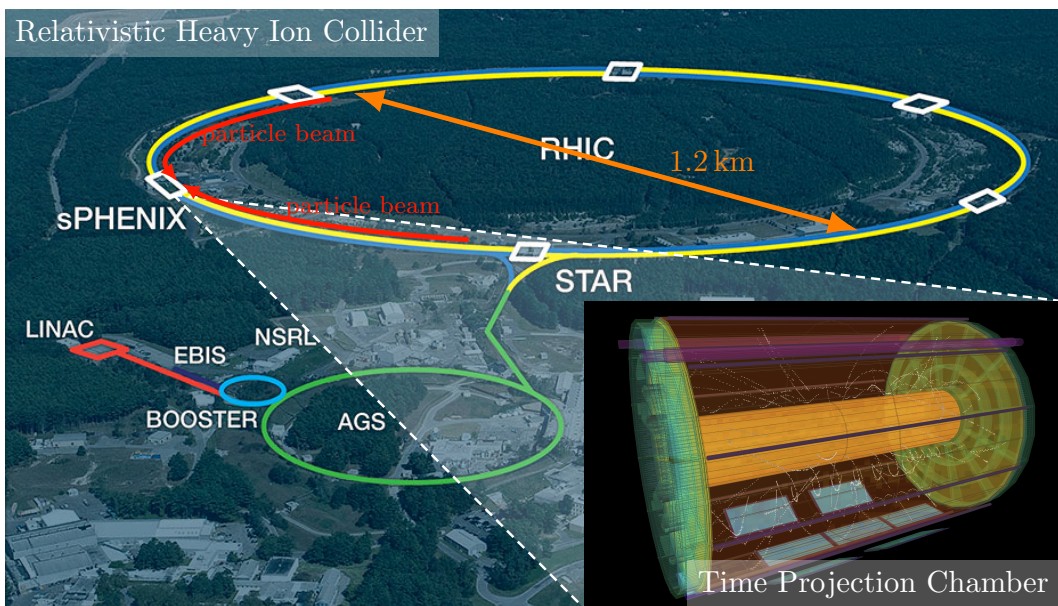

Figure 6: Relativistic Heavy Ion Collider at Brookhaven National Lab and sPHENIX Experiment.

### A.1    SIMULATION AND PROVENANCE

Minimum-bias p+p collisions are generated with PYTHIA-8.307 Sjöstrand et al. (2015) "Detroit" tune Aguilar et al. (2022) then propagated through a full GEANT4 Agostinelli et al. (2003) simulation of the as-built sPHENIX detector, including its detailed CAD geometry and measured 1.4T field map. The 'FTFP_BERT_HP' physics list is used for high-precision treatment of neutron and hadron interactions. The simulation chain models continuous energy loss, multiple scattering, secondary particle production, and decay processes with the true material budget, as well as supports space-charge distortion and its data-driven correction and carries signals through the full front-end electronics (shaping, digitization, zero suppression, and channel-by-channel gain/noise).

The simulated TPC response, so-called G4HITS, emulates raw ionization signals from charged particles traversing the TPC volume, which are reconstructed into spacepoints reflecting the true spatial resolution and distortions. Each spacepoint is then matched to the Monte Carlo truth particle that produced it, and the particle's properties – identity, momentum, and track association – are recorded as ground-truth labels for our downstream tasks (track finding, particle identification (PID), and noise identification).

### A.2    CONTENTS AND STRUCTURE

Each event contains:

- Reconstructed spacepoints from the TPC, including position and ionization energy.
- Monte Carlo truth particles with their PDG identity, momentum at production, and vertex location at production.
- Associations between spacepoints and truth particles.

## A.3 DATASET STATISTICS

The event-level complexity in the dataset varies widely. As shown in Figure 7, the number of reconstructed TPC spacepoints per event ranges from a few hundred to tens of thousands, reflecting low-multiplicity to relatively busy collision topologies. Correspondingly, the number of truth tracks per event spans from under 10 up to nearly 100.

Figure 8 summarizes the class composition for the noise-tagging and PID downstream tasks.

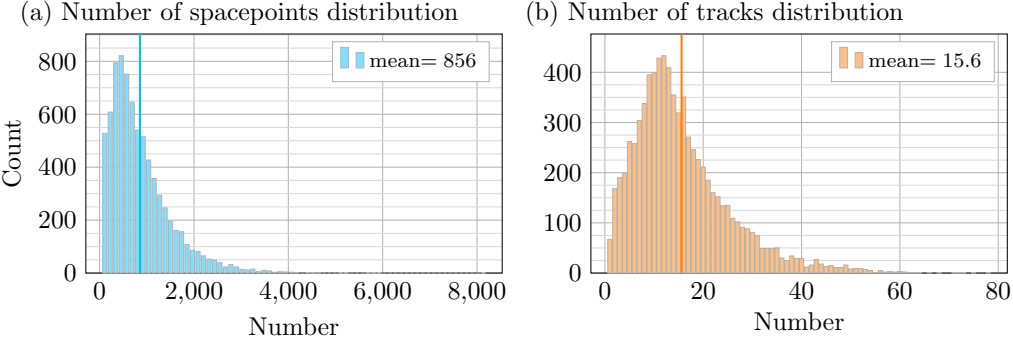

Figure 7: Distributions of number of spacepoints (a) and tracks (b) per event.

**Noise-tagging.** Noise spacepoints are defined operationally based on the truth-level kinematics of their progenitor particles. Specifically, any spacepoint associated with a Monte Carlo truth track whose momentum is below $60\ \mathrm{MeV}/c$ is labeled as *noise*. Particles produced in the primary p+p collision with such low momentum are kinematically unable to reach the active TPC volume due to the magnetic field. Therefore, spacepoints matched to these low-momentum tracks arise predominantly from secondary interactions with detector material (e.g., delta electrons, conversion products, or other material-induced processes). These secondary-origin spacepoints are not part of the primary signal topology of interest and are treated as noise for the purposes of the corresponding downstream classification task.

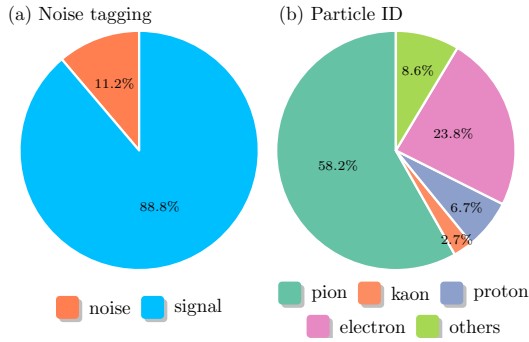

Figure 8: Class ratio of noise tagging (a) and particle identification (b).

**PID.** The PID task uses five coarse-grained target classes, grouping charge-conjugate species together to reduce sparsity while preserving physics relevance:

- **Pion:** $\pi^+$, $\pi^-$
- **Kaon:** $K^+$, $K^-$
- **Proton:** proton and anti-proton
- **Electron:** electron and positron
- **Other:** all remaining particle species.

The class ratios shown in Figure 8 reflect the inherent imbalance in these labels, driven by the underlying particle production spectra and the noise definition.

## B  METHODOLOGY

### B.1  PRELIMINARIES

This section outlines a compact mathematical way to express the hierarchical relationship between events, tracks, and spacepoints in a particle detector, such as a Time Projection Chamber (TPC). A collision **event** $E$ is represented as a set of **tracks** $\{T_j\}$, where each track $T_j$ is an ordered sequence of **spacepoints** $(s_k)$ and each spacepoint $s$ is a vector $(E_{\text{dep}}, x, y, z, \dots)$ containing its physical properties. Concretely, we express a single event, $E$, as follows:

$$E = \{T_j\}_{j=1}^m$$

This states that an event ($E$) is a set containing $m$ individual tracks ($T_j$). The number of tracks, $m$, varies for each event. Each track, in turn, is defined by its constituent spacepoints:

$$T_j = (s_{j,k})_{k=1}^{n_j}$$

This expresses that a single **track** ($T_j$) is an ordered sequence of $n_j$ spacepoints ($s_{j,k}$). The sequence is ordered because particles follow a specific path through the detector, and the number of space-points per track, $n_j$, is also variable. Finally, each individual spacepoint is a vector of its properties, which can be represented abstractly as:

$$s_{j,k} \in \mathbb{R}^D$$

A **spacepoint** ($s$) is a vector in a D-dimensional feature space. A **Spacepoint** ($s_{j,k}$) is now explicitly defined as a vector containing its primary physical properties:

$$s_{j,k} = (\mathcal{E}, x, y, z)_{j,k}$$

where $\mathcal{E}$ is the energy deposited by the particle at that point in the detector and $(x, y, z)$ is the spatial coordinates of the spacepoint.

### B.2  COORDINATE TRANSFORMATION

We transform spacepoint coordinates from Cartesian $(x, y, z)$ to a cylindrical-polar system $(r, \phi, \eta)$ that better reflects the geometry and symmetries of collider experiments. The radial distance $r$ is defined as $r = \sqrt{x^2 + y^2}$, measuring how far a point lies from the beamline in the transverse plane, and is essential for evaluating transverse momentum and energy. The azimuthal angle $\phi$ is given by $\phi = \text{atan2}(y, x)$, describing the orientation of the spacepoint in the $x$-$y$ plane and exploiting the detector's cylindrical symmetry around the beam axis. The pseudorapidity $\eta$ is defined as $\eta = -\ln[\tan(\theta/2)]$, where $\theta = \text{atan2}(r, z)$ is the polar angle. This coordinate is used instead of $\theta$ because particle production tends to be uniform in $\eta$, and, for highly relativistic particles, $\eta$ approximates the Lorentz-invariant rapidity. Finally, to ensure consistent feature scaling, we apply a min-max normalization to the spatial coordinates, transforming the pseudorapidity ($\eta \in [-2, 2]$), azimuthal angle ($\phi \in [-\pi, \pi]$), and radial distance ($r \in [30, 78]$, centimeters) into the interval $[0, 1]$. The transformed $s_i = (\mathcal{E}, r, \phi, \eta)_i$ are used for all analyses described in this paper.

### B.3  SERIALIZATION

Our objective is to perform self-supervised pretraining on the raw three-dimensional (3D) point cloud of particle spacepoints from a collision event, $S = \{s_1, \dots, s_N\}$. To leverage the power of sequential models like MAMBA, which have excelled in learning rich representations, we must first solve the fundamental problem of transforming the unordered 3D set into an ordered one-dimensional sequence. This **serialization** process is not merely a technical step. The choice of ordering scheme is critical for preserving the data's underlying physical structure.

An ideal serialization must satisfy two competing demands: it must respect the *global* physics of the event (i.e., particles flying outwards) while simultaneously preserving the *local* continuity of individual particle tracks.

We first analyze and dismiss naive approaches. For example, a space-filling curve excels at preserving 3D locality but completely disregards the concept of a track. Its path erratically jumps between physically distinct trajectories, creating a chaotic signal. Conversely, a simple global raster scan on the spacepoints' cylindrical coordinates, $s_i' = (r_i, \phi_i, \eta_i)$, respects the outward propagation along the radius but fails on local continuity. The initial hits of a track (at low $r$) become "context-starved" as their preceding elements in the sequence belong to entirely different tracks.

**Proposed Solution: Hierarchical Raster Scan**   To resolve this dichotomy, we introduce a **Hierarchical Raster Scan**. This method balances global structure with local context by operating on two levels:

1. **Partitioning:** The entire detector volume is partitioned into a grid of smaller 3D "boxes."

2. **Ordering:** A raster scan using the physically motivated order $(r, \phi, \eta)$ is applied twice. First, it orders the spacepoints *within* each box (intra-box ordering). Second, it orders the boxes themselves based on their geometric centers (inter-box ordering).

This strategy ensures the sequence progresses globally outwards but maintains local contiguity within each partitioned region. However, even with this optimal serialization, a profound challenge remains. If the learning objective simply is to predict the next hit in this sequence, the model would be forced to learn the arbitrary artifacts of the serialization itself, particularly the artificial jumps at box boundaries.

Therefore, designing a robust serialization scheme is a necessary but insufficient step. The learning objective must be intelligently designed to be independent of these serialization artifacts, a challenge we address in the subsequent section.

**Physics-informed Partitioning**   The division of the detector volume into a grid is not uniform. Instead, it is a physics-informed partitioning designed to align with both the detector's physical geometry and the observed distribution of particle hits. This ensures the partitioning itself provides a meaningful structural prior for the learning task.

For the azimuthal angle ($\phi$) and pseudorapidity ($\eta$) dimensions, the binning is data-driven. The boundaries are specifically chosen to create bins with a roughly uniform density of hits. This strategy balances the information content across partitions, preventing high-occupancy regions from disproportionately influencing the model. A detailed number of bins and illustration of this binning strategy is provided in Figure 9.

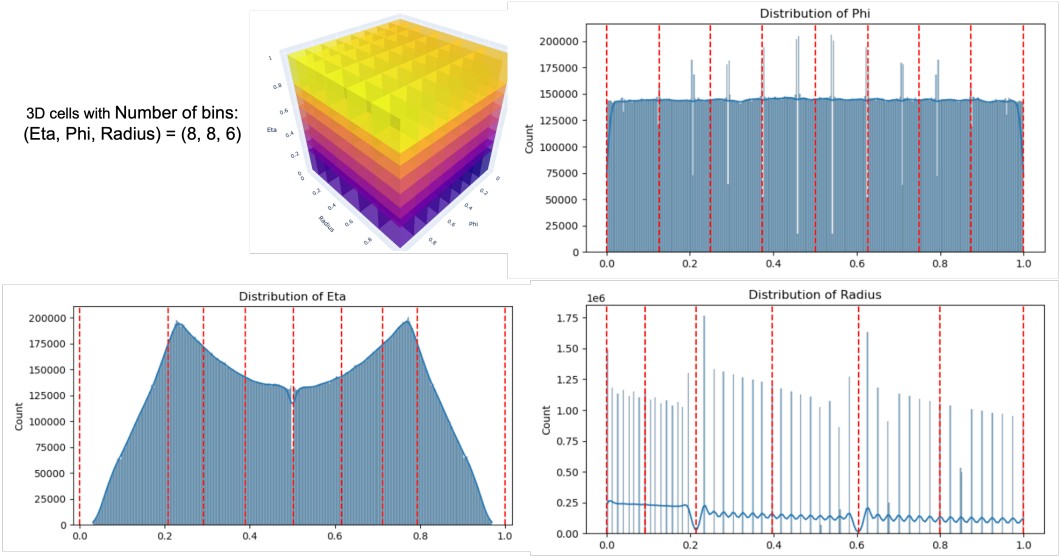

Figure 9: Physics-informed Partitioning. Top-left graph shows the binning of the data space into 384 bins ($8 \times 8 \times 6$). The other plots show the distribution of spacepoint values in normalized Phi, Eta, and Radius dimensions, respectively, computed using 50,000 events.

For the radial dimension ($r$), the partitioning mirrors the sPHENIX detector's physical construction. The detector's 48 layers are arranged in three major groups. Thus, we create six radial bins, allocating two bins to each major detector group. By embedding the detector's known layered structure into the partitioning scheme, we further ground the serialization process in the experiment's physical reality.

### B.4 MAMBA: SELECTIVE STATE SPACE MODELS

Mamba represents a significant advancement in sequence modeling, challenging the dominance of the Transformer architecture, particularly for long sequences. It is a selective state space model (SSM) that combines the strengths of recurrent neural networks (RNNs) and convolutional neural networks (CNNs) to offer linear-time complexity and constant-time inference.

Mamba's foundation is the SSM, a continuous-time system described by the following linear ordinary differential equation:

$$\frac{dh(t)}{dt} = Ah(t) + Bx(t),$$
$$y(t) = Ch(t) + Dx(t)$$

Here, $h(t)$ is the latent state, $x(t)$ is the input, and $y(t)$ is the output. $A$, $B$, $C$, and $D$ are matrices that are typically learned from data.

For use in deep learning, this continuous system is discretized. A crucial step in Mamba is making the key matrices, particularly the transition matrix $A$ and the input projection matrix $B$, selective and input-dependent. This is achieved by having dedicated neural networks that predict these matrices based on the current input token.

The discretized formulation of the state transition is:

$$h_t = \bar{A}h_{t-1} + \bar{B}x_t \tag{1}$$

Where $\bar{A}$ and $\bar{B}$ are the discretized, input-dependent matrices. This selectivity allows Mamba to modulate its recurrent state, effectively controlling how much of the past to retain and how to incorporate the current input. The model can be unrolled for efficient parallel training, similar to a CNN, or used in a recurrent manner for constant-time inference.

**Architectural Principles** The core innovation of Mamba is its selective mechanism, which allows the model to dynamically adapt its parameters based on the input. This enables it to focus on relevant information and filter out noise, a crucial capability for processing long and complex sequences. Unlike traditional SSMs, which are time-invariant, Mamba's parameters are functions of the input, making it a time-varying system. Key components of the Mamba architecture include:

- Selective State Space Layer: This is the fundamental building block of Mamba. It replaces the attention mechanism and feed-forward network of a Transformer block.

- Hardware-Aware Algorithm: Mamba employs a parallel scan algorithm that is optimized for modern hardware (GPUs), enabling efficient training and inference. This algorithm avoids the materialization of the full state sequence, a significant memory bottleneck in traditional SSMs.

**Mamba-2** Mamba-2 is a direct successor to Mamba, designed to further improve upon its efficiency and performance. It introduces a new theoretical framework called *Structured State Space Duality* (SSD), which provides a deeper understanding of the relationship between SSMs and other architectures like Transformers. The primary motivation behind Mamba-2 was to address some of the hardware utilization inefficiencies of the original Mamba. While Mamba offered linear-time complexity, its performance on modern GPUs could still be optimized. Key improvements in Mamba-2 include:

- SSD: This framework establishes a formal equivalence between a class of structured SSMs and a form of global convolution. This duality allows for the design of more efficient algorithms by leveraging insights from both perspectives.

- Architectural Simplifications: Mamba-2 simplifies the Mamba block by replacing the complex selective scan with a more structured and hardware-friendly formulation derived from the SSD framework. This often involves a multi-headed Mamba block, analogous to the multi-head attention in Transformers.

- Improved Hardware Utilization: The redesigned architecture of Mamba-2 is more amenable to parallelization on modern hardware, leading to significant speedups in both training and inference compared to the original Mamba.

### B.5 FM4NPP: Architecture

**Positional Embedding**  The model first transforms the raw input data into a high-dimensional space suitable for sequence processing. An input batch of serialized collision events is represented as a tensor of shape $(B, S, 4)$, where B is the batch size, S is the sequence length, and each space-point is a four-dimensional (4D) vector comprising its deposited energy and 3D spatial coordinates $(E_{\text{dep}}, \eta, \phi, r)$. This tensor is processed by an embedding module that projects the 4D spacepoint features into the model's latent space, $D_{\text{model}}$. It also computes a positional encoding from the 3D spatial coordinates using a function $\gamma(\cdot)$ inspired by Neural Radiance Fields (NeRF), defined as:

$$\gamma(\mathbf{p}) = \left(\mathbf{p}, \sin(2\mathbf{p}), \cos(2\mathbf{p}), \ldots, \sin\left(2^l\mathbf{p}\right), \cos\left(2^l\mathbf{p}\right)\right)$$

where $\mathbf{p}$ is the coordinate vector and the frequencies $2^l$ are sampled from a geometric progression. This encoding, also mapped to $D_{\text{model}}$, is combined with the feature representation via element-wise addition. The output of this stage is a single tensor of shape $(B, S, D_{\text{model}})$, where $D_{\text{model}}$ is the model width.

**Network Architecture and k-Next Nearest Neighbor Prediction Head**  The core architecture consists of a stack of Mamba blocks that sequentially process the embedded hits. The input to the first block is the $(B, S, D_{\text{model}})$ tensor from the embedding stage. Each block operates as follows:

- Pre-Normalization: The input tensor is first passed through a Root Mean Square Normalization (RMSNorm) layer. This layer normalizes the feature vector of each spacepoint independently.

- Sequence Modeling: The normalized $(B, S, D_{\text{model}})$ tensor is then processed by the Mamba-2 layer.

- Residual Connection: A residual or "skip" connection is applied around the normalization and Mamba-2 layers. The original input to the block is added element-wise to the output of the Mamba-2 layer.

After passing through the final Mamba block, the sequence is processed by one last RMSNorm layer. The resulting $(B, S, D_{\text{model}})$ tensor is then fed into the prediction head. This head is a single linear layer that projects the $D_{\text{model}}$-dimensional representation of each hit to a $3k$-dimensional vector, yielding a final output tensor of shape $(B, S, 3k)$. Here, $k = 30$ is the number of neighbors to be predicted. This output format is designed specifically for the Causal k-Nearest Neighbor (kNN) objective.

### B.6 Maximal Update Parameterization

**Challenge in Scaling Models**  Imagine building with LEGOs. If you build a small car, it is stable. Yet, if you try to build a life-sized car using the exact same small-brick techniques, it will be flimsy and fall apart. Modern AI models face a similar problem. When we try to make them bigger and more powerful by adding more "width" or digital neurons, their internal mathematics can become unstable during training. The signals inside can either "explode" into uselessly large numbers or "vanish" to zero, making it impossible for the model to learn. μ-Parameterization (μP) is a groundbreaking set of rules that solves this problem. It is much like a master blueprint for building AI models, explaining exactly how to adjust the initial settings and learning rate based on the model's size. This ensures that as the model scales up, its internal signals stay perfectly balanced, allowing it to train stably and effectively. A major benefit is the best training settings found on a small, cheap model can be directly transferred to a massive, expensive one, saving enormous amounts of time and computational cost.

Concretely, standard infinite-width network analyses, such as those based on the Neural Tangent Kernel (NTK), predict that wide networks operate in a "lazy regime," where they fail to learn meaningful features from data. μP was introduced to overcome this limitation by defining a specific scaling of model initializations and learning rates that guarantee nontrivial feature evolution in the infinite-width limit. A significant practical advantage of μP is that it enables zero-shot hyperparameter transfer, allowing optimal settings found on small-scale models to be directly applied to their large-scale counterparts. This mitigates the often prohibitive computational costs associated with tuning large models.

**Applications in Modern Architectures**    The principles of μP have been successfully extended beyond simple multilayer perceptrons (MLPs) to a range of complex architectures. In Transformers, μP facilitates hyperparameter transfer, although achieving a stable feature-learning limit requires careful scaling with respect to both model width and depth. The framework also has been adapted for scientific machine learning models, such as Fourier Neural Operators (FNOs), where a specific μ-FNO parameterization ensures stable training as the model size and number of Fourier modes are scaled. Recently, μP has been applied to stabilize training of large Diffusion Models, again enabling hyperparameter transfer for these computationally intensive generative systems. This body of research highlights both the generality of the μP framework and the necessity of deriving architecture-specific scaling laws.

**μP for MAMBA**    To address this, a corrected scaling for SSMs, termed $\mu$P-SSM (*Maximal Update Parameterization for SSMs*), was derived by analyzing signal propagation directly within the Mamba architecture. This analysis yielded specific scaling rules for initialization variances ($\sigma$), which control the scale of the model's initial random weights, and learning rates ($\eta$) that determine the step size during training. The key formulas dictate how these parameters for Mamba's weight matrices ($W_B, W_C$) should be scaled relative to the model's latent state dimension ($N_x$) and input dimension ($N_u$). Using asymptotic Big-Theta ($\Theta$) notation, the rules are:

- **Initialization Variances:** $\sigma_B \in \Theta(\sqrt{\frac{N_x}{N_u}})$ and $\sigma_C \in \Theta(\frac{1}{\sqrt{N_x N_u}})$

- **Learning Rates:** $\eta_B \in \Theta(\frac{N_x}{\sqrt{N_u}})$ and $\eta_C \in \Theta(\frac{1}{N_x \sqrt{N_u}})$

We have integrated this $\mu$P-SSM methodology into our own Mamba-based model. The effectiveness of this approach is evidenced by the stable scaling of layer-wise activation norms across different model sizes as empirically verified in our experiments. Unlike standard parameterizations that lead to exploding signals or heuristic $\mu$P which leads to vanishing signals, our model's activations and their updates remain correctly scaled, confirming that the model is operating in a stable feature-learning regime.

### B.7    ADDITIONAL DETAILS FOR PRETRAINING

The model is trained using the AdamW optimizer, which incorporates weight decay for regularization against overfitting. To manage the learning rate dynamics, we employ a cosine decay schedule, which is preceded by a brief linear warmup period at the beginning of training to ensure initial stability. To further prevent training instabilities arising from large gradients, we apply gradient clipping. The learning objective is to minimize a Mean Squared Error (MSE) loss function. This loss quantifies the Euclidean distance between the model's predicted coordinates for the kNN and the truth coordinates. These truth neighbors are pre-computed for each particle spacepoint during the data loading phase to ensure efficient throughput during training.

**Loss Rescaling by Event Difficulty**    We identified a nuisance structure in the training data related to event spacepoint density: events with a larger number of spacepoints are inherently easier to predict as the average distance between neighboring spacepoints is smaller. This variance in difficulty can lead to training instability, manifesting as loss spikes. To mitigate this, we introduce a loss rescaling strategy based on event binning. Events are first grouped into discrete bins based on their average kNN distance, which serves as a proxy for prediction difficulty. Let $g(i)$ be the function that maps event $i$ to its corresponding difficulty bin. The loss objective is then modified as follows: 1) the MSE for each event is re-weighted by a factor $w_{g(i)}$, corresponding to the average difficulty of its bin, and 2) the total batch loss is calculated by averaging these re-weighted individual losses. This is formulated as:

$$\mathcal{L} = \frac{1}{B} \sum_{i=1}^{B} w_{g(i)} \mathcal{L}_i = \frac{1}{B} \sum_{i=1}^{B} w_{g(i)} \left( \frac{1}{S_n} \sum_{j=1}^{S_n} ||\mathbf{s}_{ij} - \mathbf{y}_{ij}||_2^2 \right)$$

Here, $B$ is the number of events in the batch; $\mathcal{L}_i$ is the standard MSE for event $i$ with $S_n$ spacepoints; $\mathbf{s}_{ij}$ and $\mathbf{y}_{ij}$ are the predicted and truth coordinates, respectively; and $w_{g(i)}$ is the pre-computed weight for the difficulty bin to which the event belongs. This ensures a single batch-averaged loss is computed only after accounting for the inherent difficulty of each event in the batch.

## C    ADDITIONAL RESULTS

### C.1    DOWNSTREAM MODEL

#### C.1.1    TRACKING (INSTANCE SEGMENTATION)

Our lightweight downstream model for track finding – formulated as a per-point instance segmentation task – is inspired by image panoptic segmentation models, such as MASKFORMER and MASK2FORMER, adapted to point cloud data.

Let $\mathbf{X} = \{\mathbf{x}_i\}_{i=1}^N$ denote the input set of $N$ points, where each $\mathbf{x}_i \in \mathbb{R}^d$ is a $d$-dimensional point-level feature (either raw input, pretrained representation, or from a randomly initialized encoder). These are first projected into a latent embedding space via a linear layer:

$$\mathbf{e}_i = \mathbf{W}_{\text{proj}}\mathbf{x}_i, \quad \mathbf{e}_i \in \mathbb{R}^{d_e}.$$

We denote the set of projected spacepoint embeddings as $\mathbf{E} = \{\mathbf{e}_i\}_{i=1}^N$.

To represent candidate tracks, we use $K$ learnable queries (track queries) $\mathbf{Q}^{(0)} = \{\mathbf{q}_k^{(0)}\}_{k=1}^K$, where each $\mathbf{q}_k^{(0)} \in \mathbb{R}^{d_e}$. These prototypes are refined over $L$ transformer decoder layers. Each decoder layer consists of:

- **Cross-attention:** updates $\mathbf{q}_k$ by attending to point embeddings $\mathbf{E}$.
- **Self-attention:** refines interaction among the $K$ prototypes.
- **Feed-forward network (FFN):** standard Transformer update.

After $L$ decoder layers, we obtain the refined track queries $\mathbf{Q}^{(L)} = \{\mathbf{q}_k^{(L)}\}_{k=1}^K$. Each refined query vector is then processed by two MLPs:

$$\mathbf{m}_k = \text{MLP}_{\text{mask}}(\mathbf{q}_k^{(L)}), \quad \hat{y}_k = \text{MLP}_{\text{cls}}(\mathbf{q}_k^{(L)}),$$

where $\mathbf{m}_k \in \mathbb{R}^{d_e}$ is the track embedding for the $k$-th prototype and track instance prediction $\hat{y}_k \in [0, 1]$ is the probability of corresponding to a real track (versus a "no-object" class).

Each track embedding $\mathbf{m}_k$ is used to compute point-to-prototype assignment logits:

$$z_{ik} = \mathbf{e}_i^\top \mathbf{m}_k, \quad \hat{p}_{ik} = \sigma(z_{ik}),$$

where $\sigma(\cdot)$ denotes the sigmoid function. The predicted assignment probability $\hat{p}_{ik}$ represents the likelihood that point $i$ belongs to prototype $k$.

To encourage each track query to focus on the subset of points it is likely responsible for, we apply an *additive attention mask* during cross-attention. The attention mask is defined as:

$$A_{ik} = -\log(\hat{p}_{ik} + \epsilon)$$

with a small constant $\epsilon$ added for numerical stability. This mask is added to the attention logits before the softmax operation in the cross-attention layer. This dynamic masking suppresses contributions from low-probability points and improves localization by making each prototype attend selectively to its likely constituent points.

**Training Loss.**    Let $\mathcal{T} = \{T_j\}_{j=1}^M$ be the set of $M$ ground-truth tracks (instance labels). We compute a bipartite matching between the $M$ ground-truth tracks and the $K$ refined track queries using the Hungarian algorithm. The matching minimizes a cost function combining:

- Dice loss $\mathcal{L}_{\text{dice}}$ on the per-point predicted versus ground-truth track
- Focal loss $\mathcal{L}_{\text{focal}}$ on point-wise assignment probabilities
- Classification loss $\mathcal{L}_{\text{cls}}$ on the track/no-object prediction.

For each matched pair $(T_j, \mathbf{q}_k)$, the total loss is:

$$\mathcal{L}_{\text{match}}^{(j,k)} = \lambda_{\text{dice}} \cdot \mathcal{L}_{\text{dice}}^{(j,k)} + \lambda_{\text{focal}} \cdot \mathcal{L}_{\text{focal}}^{(j,k)} + \lambda_{\text{cls}} \cdot \mathcal{L}_{\text{cls}}^{(k)}.$$

For unmatched prototypes, we only compute $\mathcal{L}_{\text{cls}}^{(k)}$ with the ground truth label being "no-object."

The final training loss includes auxiliary losses from each decoder layer $\ell = 1, \ldots, L$, as well as from the initial prototype vectors:

$$\mathcal{L}_{\text{total}} = \sum_{\ell=0}^{L} \mathcal{L}^{(\ell)}.$$

During inference, we assign each spacepoint $i$ to the track whose combined mask and classification score is maximal. Concretely, we compute $k_i^* = \arg\max_k \left( \hat{p}_{ik} \, \hat{y}_k \right)$ and label point $i$ as belonging to track $k_i^*$.

This formulation enables end-to-end training of the instance segmentation model, while allowing the pretrained or learned point embeddings to guide track-level grouping.

### C.1.2 Particle Identification and Noise Identification

For both PID and noise classification, we use a simple lightweight adapter:

- **Embedding:** A linear layer projects each point feature $\mathbf{x}_i \in \mathbb{R}^d$ to a $d_p$-dimensional embedding.
- **Context:** A single self-attention layer aggregates global information across all point embeddings.
- **Prediction:** An MLP with softmax over $C$ output classes.

## C.2 Comparative Methods for Downstream Tasks

### C.2.1 Adapt Exa.TrkX Pipeline for sPHENIX Tracking-Finding

In this section, we discuss several adaptions done to the Exa.TrkX pipeline for it to work well on the sPHENIX data. We need to apply adaptions to the first four stages – data pre-processing, hit embedding, edge filtering, and GNN edge classification – out of six stages of the Exa.TrkX pipeline.

**Pre-processing.** The Exa.TrkX's study was based on the TrackML dataset Amrouche et al. (2020). The dataset provides two sources for the construction of the neural network input – the 3D location of the spacepoints and the directional information and summary statistics from the charge deposited in each spacepoint (eight-dimensional). The second source of information is called *cell features* in the paper. The hit feature is the concatenation of the location and cell features. Because sPHENIX data do not provide cell features, we only used the location of hits in the HEP-coordinate to construct the input. More precisely, let $(\hat{\eta}, \phi, \hat{r})$ be the location of a hit (normalized pseudorapidity, angle, and normalized radius), the features of this hit is a five-dimensional vector

$$(\mathcal{E}, \hat{\eta}, \cos(\phi), \sin(\phi), \hat{r}),$$

where $\mathcal{E}$ is the energy. We use $(\cos(\phi), \sin\phi)$ instead of $\phi$ to overcome the discontinuity of $\phi$ at $2\pi$. We normalize the pseudorapidity $\eta$ by 1.96 to get the normalized pseudorapidity $\hat{\eta} \in (-1, 1)$. To normalize a radius, we first match it to the closest one of the 48 radius bins and use the bin index to replace the radius. Then, we divide the index by 48 to normalize the radius to a number between $[0, 1)$. We do this because the distance between the sPHENIX TPC layers are not uniform with outer layers, spacing farther apart than the inner ones. This may be a problem for distance-based edge set construction for a GNN model as same-track hits toward the end of the track may be less likely to be connected by the model.

**Embedding and Filtering.** The Exa.TrkX pipeline embeds the spacepoints and filters the edges as two separate steps. To adapt them for sPHENIX, we modified the procedure in the following aspects: 1) how to determine whether a pair of hits is connected, 2) how candidate hit pairs are generated, 3) how to traine the models, and 4) how to construct the models.

In the embedding stage, Exa.TrkX trained an MLP network to embed each hit into a latent representation, so pairs of neighboring hits from the same track are closer in the latent space than pairs that are not (e.g., from different tracks or not neighbors on the same track). The embedding network is trained by first passing the two hits through the same embedding network then minimizing the hinge loss of the distance between the two embeddings.

Because sPHENIX data do not provide information to determine if two same-track hits are direct neighbors (although this information could be inferred for high-energy tracks), we decide not to

distinguish whether two same-track hits are neighboring. This approach also was recommended by the `Exa.TrkX` research team as a valid alternative.

In the filtering stage, `Exa.TrkX` takes a pair of hits, passes them through the embedding network, concatenates the two embeddings, and passes the concatenation through a MLP filtering network to predict if the two hits are connected. The prediction is optimized by a binary cross entropy loss.

For both the embedding and filtering models, we need to provide candidate hit pairs. For the embedding stage, `Exa.TrkX` uses two types of pairs, random and kNN pairs, as a form of hard negative mining. As a random pair has an extremely low chance to be connected, `Exa.TrkX` also trains on pairs formed by a hit and its closest neighbors in the latent representation space.

We follow the pipeline as closely as possible. However, because of the differences between `sPHENIX` and `TrackML` input features and the fact we treat all pairs from the same track as being connected (in contrast to `Exa.TrkX`'s approach where only immediate neighbors are connected), we have to choose different cutoffs in both embedding and filtering. Specifically, we set a threshold of 2. for distance in the embedding space with pairs less than the threshold apart classified as having an edge between them. The threshold was chosen as it ensures we have an over .8 recall (efficiency in the `Exa.TrkX` terminology) in identifying pairs from the same track. Of note, here we did not select a threshold that ensures close to a $100\%$ recall because we can afford the model to fail to recognize faraway points from the same track as being connected.

For the filtering step, we choose a threshold of .675 for probability of a true edge with pairs exceeding the threshold considered as being connected. The threshold is selected because it ensures the false positive rate in edge identification will dip below $1\%$.

**GNN edge classification.** For the final GNN step, we also employ the Interaction Network Battaglia et al. (2016) architecture with the same hyperparameters used by the `Exa.TrkX` pipeline. For edge classification, we choose a threshold of .9 as probability of a true edge. With this choice, we achieved a $91.79\%$ tracking efficiency (recall) ($94.74\%$ for tracks with $p_\mathrm{T} > 1\,\mathrm{GeV}$) and a track purity (precision) of $66.42\%$. With a threshold of .8, the tracking efficiency drops slightly to $90.01\%$ ($92.60\%$ for $p_\mathrm{T} > 1\,\mathrm{GeV}$) with a notable improvement in purity to $76.72\%$.

### C.2.2 ADAPT EGGNET FOR SPHENIX TRACK-FINDING

The `EggNet` study also is based on the `TrackML` dataset Amrouche et al. (2020) and shares the same data pre-processing approach with `Exa.TrkX`. To partially compensate for the lack of cell features from `sPHENIX` data, we use the following approach to augment the input. Let $(\hat{\eta}_0, \phi_0, \hat{r}_0)$ be the location of a hit (normalized pseudorapidity, angle, and normalized radius). The features of this hit is a 12-dimensional vector

$$(\hat{\eta}_0, \cos(\phi_0), \sin(\phi_0), \hat{r}_0;\ \hat{\eta}_1, \cos(\phi_1), \sin(\phi_1), \hat{r}_1;\ \hat{\eta}_2, \cos(\phi_2), \sin(\phi_2), \hat{r}_2)\,,$$

where $(\hat{\eta}_1, \phi_1, \hat{r}_1)$ and $(\hat{\eta}_2, \phi_2, \hat{r}_2)$ are the locations of the two closest neighbors of the hit in the $(\hat{\eta}, \cos(\phi), \sin(\phi), \hat{r})$ space. The motivation for augmenting the hit with the two closest neighbors is that for the majority of the hits in a high-energy track, the two closest neighbors are most likely from the same track. In which case, the augmented hit features can provide information about the track's direction.

For the GNN model, `EggNet` adopted a similar approach to `GravNet` Qasim et al. (2019). The outstanding feature of a `GravNet`-type model is that the edge set is not predetermined but constructed dynamically. More precisely, `EggNet` will run $N$ normal GNN iterations, but, before each GNN iteration, the edge set will be constructed via kNN based on the current node embeddings. To adapt `EggNet` to `sPHENIX` data, we set GNN iterations to be 4 and used four message-passing rounds for each GNN iteration. The nearest 10 hits in the embedding space are used to form the neighborhood of a hit. Different from the original `GravNet` (but similar to the interaction GNN used by `Exa.TrkX`), `EggNet` also has an edge network for calculating edge messages. Moreover, `EggNet` also uses a dedicated node decoding network to produce the node embeddings for the kNN. All sub-networks of `EggNet` (node encoding/decoding networks and edge network) are MLPs, each with two hidden layers and 64 hidden features. The embedding dimension of the node (i.e., the number output features from the node decoding network) is 24.

The network is trained with a hinge loss of margin 1, aimed at reducing the Euclidean distance in the embedding space between a pair of hits from the same track and enlarging the distance between

a pair from different tracks. The model was trained for 300 epochs, and the final clustering was done using DBSCAN with $\epsilon = 1$ and minimum number samples $= 2$.

### C.2.3 ADAPT HEPT FOR SPHENIX TRACK-FINDING

`HEPT` Miao et al. (2024) is a locality-sensitive hashing-based efficient point Transformer designed for large-scale point cloud processing in high energy physics. Unlike `Exa.TrkX` and `EggNet`, which rely on graph neural networks, `HEPT` uses self-attention mechanisms with LSH-based approximation to achieve near-linear computational complexity.

To adapt `HEPT` for sPHENIX TPC tracking, we made the following modifications to the model and training procedure:

**Pre-processing.** We use the same normalized high energy physics coordinates as discussed in C.2.1: $(\hat{\eta}, \cos(\phi), \sin(\phi), \hat{r})$, where $\hat{\eta}$ is the normalized pseudorapidity and $\hat{r}$ is the normalized and binned radius. For the input features, we concatenate the energy $\mathcal{E}$ of each hit with its Cartesian coordinates $(x, y, z)$ and the normalized high energy physics coordinates, resulting in an eight-dimensional feature vector per spacepoint. Unlike some baseline methods, we do not filter tracks by transverse momentum $p_T$ and considered particles across all momentum ranges.

**Contrastive Learning.** `HEPT` is trained using a contrastive learning objective that brings embeddings of same-track hits closer together while pushing embeddings from different tracks apart. For negative sampling, we form negative examples from at most 64 neighboring hits from different tracks (reduced from the original 256 to accommodate the lower spacepoint density in sPHENIX TPC data).

**Training Configuration.** We use the most recent model architecture from the `HEPT` example folder. For optimization, we set the initial learning rate to 0.0001 (instead of the original 0.01) and switched to the AdamW optimizer (from Adam) for better regularization. These adjustments were necessary to achieve stable training convergence on the sPHENIX dataset.

**Track Formation.** Because `HEPT` produces per-point embeddings rather than end-to-end tracking predictions, we apply HDBSCAN McInnes et al. (2017) clustering on the learned embeddings to form track candidates. We use the following HDBSCAN hyperparameters: `metric="euclidean"`, `min_cluster_size=12`, `min_samples=15`, and `cluster_selection_method="eom"`. These parameters are tuned to maximize the average per-event Adjusted Rand Index (ARI) on the test set, balancing cluster granularity and noise robustness.

### C.2.4 ADAPT GNNS FOR SPHENIX PARTICLE IDENTIFICATION AND NOISE TAGGING

We selected four GNN models, `GATConv`, `GCNConv`, `GraphConv`, and `SAGEConv`, as benchmarking algorithms for the PID and noise tagging downstream tasks. We used the `torch_geometric` implementations for the models. We used the same data pre-processing protocol as discussed in C.2.1. To generate the edge set, for a hit at location $(\eta, \cos(\phi), \sin(\phi), \hat{r})$, we connect to it 50 nearest neighbor hits with distance $< 1$. We allowed the edges to be directed. The node features to the GNNs are the energy $\mathcal{E}$ of the hit together with its 4D location. For the node encoding network, we use an MLP, each with two hidden layers and 256 hidden features. We use uniformly six GNN layers for each GNN model. For the hit classification network, we use an MLP of two hidden layers with 128 and 64 hidden features. The GNNs are trained with cross entropy loss. Each GNN is trained for 200 epochs.

In general, GNNs' performance on the two downstream tasks are suboptimal. We hypothesize that the failure of GNNs is a result of their difficulty in capturing and communicating more global patterns of the tracks as solving both particle identification and noise tagging requires a model to understand the general shape of tracks that span a significant space in TPC.

### C.2.5 ADAPT ONEFORMER3D FOR SPHENIX PARTICLE IDENTIFICATION AND NOISE TAGGING

`OneFormer3D` is a state-of-the-art object detection algorithm for 3D point cloud data that can solve semantic and instance segmentation task in one run. The model architecture of `OneFormer3D` is U-Net backbone followed by a transformer decoder.

To run `OneFormer3D` on a point cloud data, we first need to get the so-called "super points" (a grouping of raw points) either by a clustering algorithm or voxelization. To adapt `OneFormer3D` to `sPHENIX` data, we use the same pre-processing approach as discussed in C.2.1 and voxelized the resulting point cloud to a grid of shape $(64, 64, 48)$ in $\hat{\eta}, \phi, \hat{r}$, respectively.

The super points first pass through the sparse convolution-powered U-Net backbone to be featurized. Then, the super point features serve as the keys and values in the Transformer encoder. The learnable queries output from the transformer decoder are then used to produce instance/semantic segmentation predictions on the super points. In the final step, the prediction on the super points will be broadcast to their constituent raw points. As both particle identification and noise tagging can be considered as semantic segmentation tasks, we separate the part of the code (primarily in prediction and loss function) for semantic segmentation from `OneFormer3D` while keeping the neural architecture identical. We use the same network parameters as the example of `OneFormer3D` on the S3DIS dataset.

Table 5: Noise tagging per-class recall and precision.

|  | Accuracy | Macro | | Non-noise | | Noise | |
|---|---|---|---|---|---|---|---|
|  |  | Recall | Precision | Recall | Precision | Recall | Precision |
| GATConv | 0.9099 | 0.6730 | 0.8060 | 0.9788 | 0.9242 | 0.3672 | 0.6878 |
| GCNConv | 0.9095 | 0.6728 | 0.8037 | 0.9784 | 0.9241 | 0.3672 | 0.6832 |
| GraphConv | 0.9190 | 0.7213 | 0.8252 | 0.9764 | 0.9351 | 0.4661 | 0.7152 |
| SAGEConv | 0.9174 | 0.7227 | 0.8165 | 0.9740 | 0.9355 | 0.4714 | 0.6975 |
| OneFormer3D | 0.9646 | 0.9404 | 0.8948 | 0.9716 | 0.9884 | 0.9092 | 0.8012 |
| AdapterOnly | 0.9111 | 0.6215 | 0.8359 | 0.9901 | 0.9169 | 0.2528 | 0.7548 |
| FM4NPP(m6) | 0.9708 | 0.9122 | 0.9114 | 0.9809 | 0.9812 | 0.8435 | 0.8416 |

Table 6: Particle Identification per-class recall and precision.

|  | Accuracy | Macro | | Others | | Pion | | Kaon | | Proton | | Electron | |
|---|---|---|---|---|---|---|---|---|---|---|---|---|---|
|  |  | Rec. | Pre. | Rec. | Pre. | Rec. | Pre. | Rec. | Pre. | Rec. | Pre. | Rec. | Pre. |
| GATConv | 0.6922 | 0.3973 | 0.6368 | 0.0947 | 0.5709 | 0.9106 | 0.7014 | 0.0057 | 0.6146 | 0.4567 | 0.6117 | 0.5190 | 0.6854 |
| GCNConv | 0.6892 | 0.3911 | 0.6319 | 0.0782 | 0.5762 | 0.9140 | 0.6966 | 0.0073 | 0.5871 | 0.4501 | 0.6140 | 0.5058 | 0.6858 |
| GraphConv | 0.7079 | 0.4176 | 0.6425 | 0.1304 | 0.5739 | 0.9133 | 0.7146 | 0.0080 | 0.5791 | 0.4766 | 0.6272 | 0.5597 | 0.7178 |
| SAGEConv | 0.7262 | 0.4563 | 0.6502 | 0.1085 | 0.5790 | 0.9126 | 0.7382 | 0.0338 | 0.5239 | 0.6242 | 0.7071 | 0.6024 | 0.7028 |
| OneFormer3D | 0.7701 | 0.4897 | 0.5767 | 0.3029 | 0.5758 | 0.9207 | 0.7658 | 0.0000 | 0.0000 | 0.4859 | 0.6991 | 0.7389 | 0.8427 |
| AdapterOnly | 0.6631 | 0.3387 | 0.6111 | 0.0095 | 0.7714 | 0.9511 | 0.6596 | 0.0002 | 0.2872 | 0.4120 | 0.6366 | 0.3209 | 0.7008 |
| FM4NPP(m6) | 0.904 | 0.6623 | 0.8328 | 0.4449 | 0.7647 | 0.9551 | 0.8484 | 0.2712 | 0.7829 | 0.8068 | 0.8763 | 0.8336 | 0.8919 |

Table 7: Diagnostic metrics for tracking performance.

| model | ARI | overall spacepoint efficiency | overall spacepoint purity | no. parameters |
|---|---|---|---|---|
| EggNet | 0.7256 | 93.01% | 92.34% | 0.16M |
| Exa.TrkX | 0.8765 | 94.47% | 98.83% | 3.86M |
| AdapterOnly | 0.7243 | 89.34% | 92.09% | 2.39M |
| FM4NPP(m6) | 0.9448 | 97.56% | 98.34% | 188M + 2.39M |

## C.3 ADDITIONAL RESULT ON SHARED VERSUS INDEPENDENT ADAPTER ARCHITECTURES FOR DOWNSTREAM TASKS

To address whether the downstream tasks can benefit from end-to-end joint training with shared adapter layers, we conduct an ablation study comparing multitask learning against isolated task-specific adapters. We focus on the two most architecturally similar tasks: PID and noise tagging, both of which operate on point-level representations.

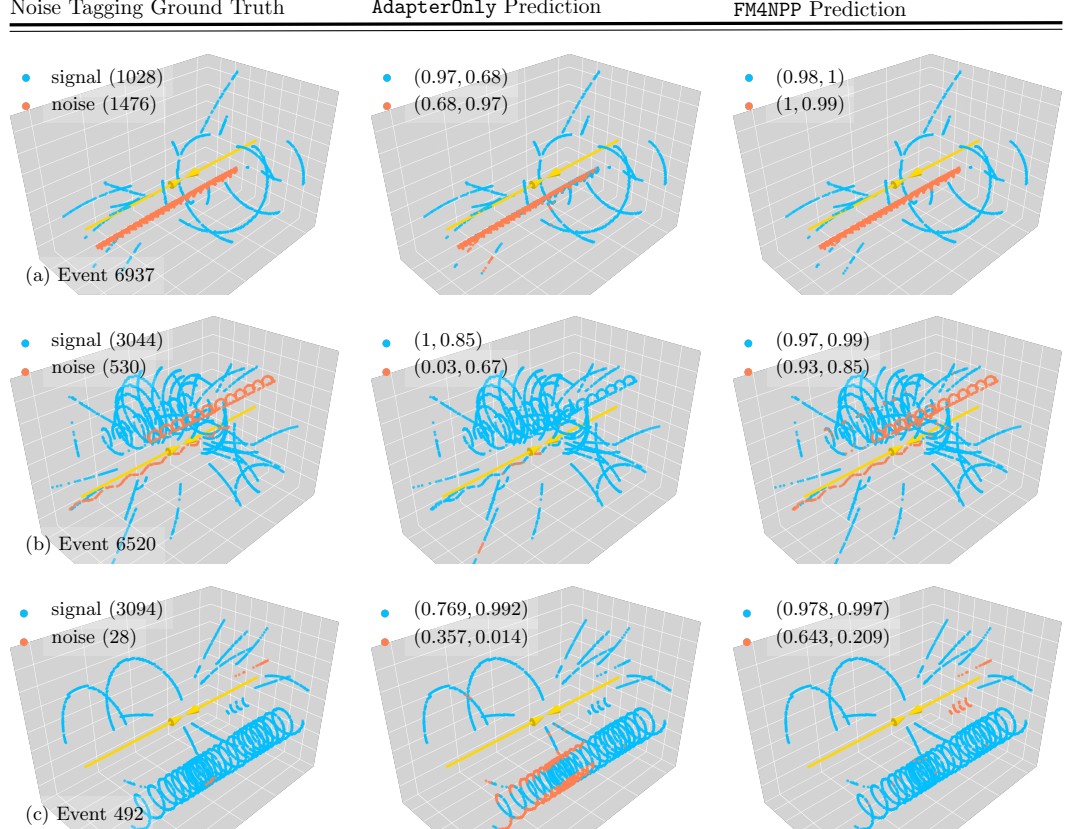

Figure 10: **Performance of `AdapterOnly` and `FM4NPP` on Noise Tagging.** The numbers in the parentheses in the target sub-figures are the number of signal and noise spacepoints. The numbers in the parentheses in the prediction sub-figures are the recall and precision of the class.

**Multitask Architecture.** The multitask model shares a common input projection layer followed by two self-attention (SA) and feed-forward network (FFN) layers across both tasks with separate task-specific classification heads for PID and NID. The training objective combines both task losses with manual weighting: $\mathcal{L}_{\text{total}} = w_{\text{PID}}\mathcal{L}_{\text{PID}} + w_{\text{NID}}\mathcal{L}_{\text{NID}}$. We use the pretrained m5 backbone (1536-dim, frozen) with weights $w_{\text{NID}} = 2.5$ and $w_{\text{PID}} = 0.5$ to prioritize the simpler binary noise tagging task.

**Results.** Table 8 compares the best validation losses achieved by multitask learning against isolated training, where each task uses its own dedicated adapter layers. Despite sharing representations through common SA+FFN layers, the multitask model exhibits *negative transfer*: both tasks perform worse than when trained independently. The NID task degrades by 7.61%, while the PID task suffers a more severe 30.69% increase in validation loss.

Table 8: Comparison of multitask versus isolated adapter training for PID and noise tagging tasks. Both configurations use the frozen m5 backbone with two shared SA+FFN layers. Lower validation loss is better.

| Task | Isolated Training | Multi-Task (Shared) | $\Delta$ (%) |
|---|---|---|---|
| Noise Tagging (NID) | 0.0513 | 0.0552 | +7.61% |
| Particle ID (PID) | 0.2377 | 0.3107 | +30.69% |

**Interpretation.** These results suggest that despite conceptual overlap between PID and noise tagging, both classify individual spacepoints, the learned representations required for optimal performance differ substantially between tasks. The negative transfer likely arises from conflicting gradient

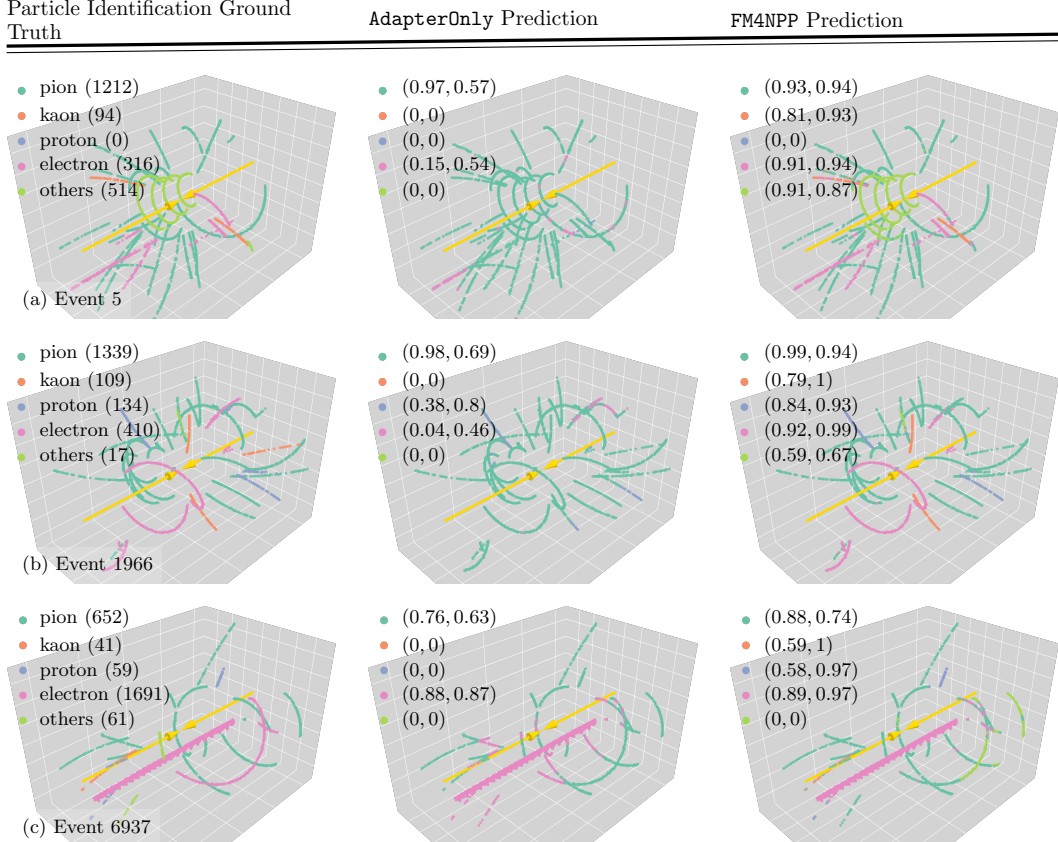

Figure 11: **Performance of `AdapterOnly` and `FM4NPP` on particle identification.** The numbers in the parentheses in the target sub-figures are the number of spacepoints in each particle ID class. The numbers in the parentheses in the prediction sub-figures are the recall and precision of the class.

signals: noise tagging requires distinguishing signal from detector noise based on energy deposition patterns, while PID must differentiate between particle species using ionization profiles. The more severe degradation in PID performance indicates the shared adapter prioritizes the simpler, more heavily weighted NID task at the expense of the more complex PID classification.

For the tracking task, which employs a fundamentally different DETR (DEtection TRansformer)-style set prediction architecture rather than point classification, joint training with PID/NID is even less suitable. Consequently, we retain independent task-specific adapters for each downstream task, allowing each to specialize its learned representations without interference while still leveraging the shared pretrained backbone.

## C.4    ADDITIONAL RESULT ON ADAPTER HEAD CAPACITY VERSUS FOUNDATION MODEL QUALITY TRADE-OFF

To understand whether adapter capacity or foundation model quality is the limiting factor for downstream performance, we conduct an ablation study on the PID task by sweeping adapter depth while keeping the pretrained `m6` backbone frozen.

**Experimental Setup.**    We varied the number of self-attention (SA) layers in the adapter head from 0 to 4 while maintaining the frozen `m6` backbone. The configurations tested include:

- **0L** (0.56M params): No SA layers, only linear projection + MLP
- **1L** (1.09M params): 1 SA layer
- **2L** (1.62M params): 2 SA layers
- **4L** (2.67M params): 4 SA layers

Track Finding Ground Truth      `AdapterOnly` Prediction      `FM4NPP` Prediction

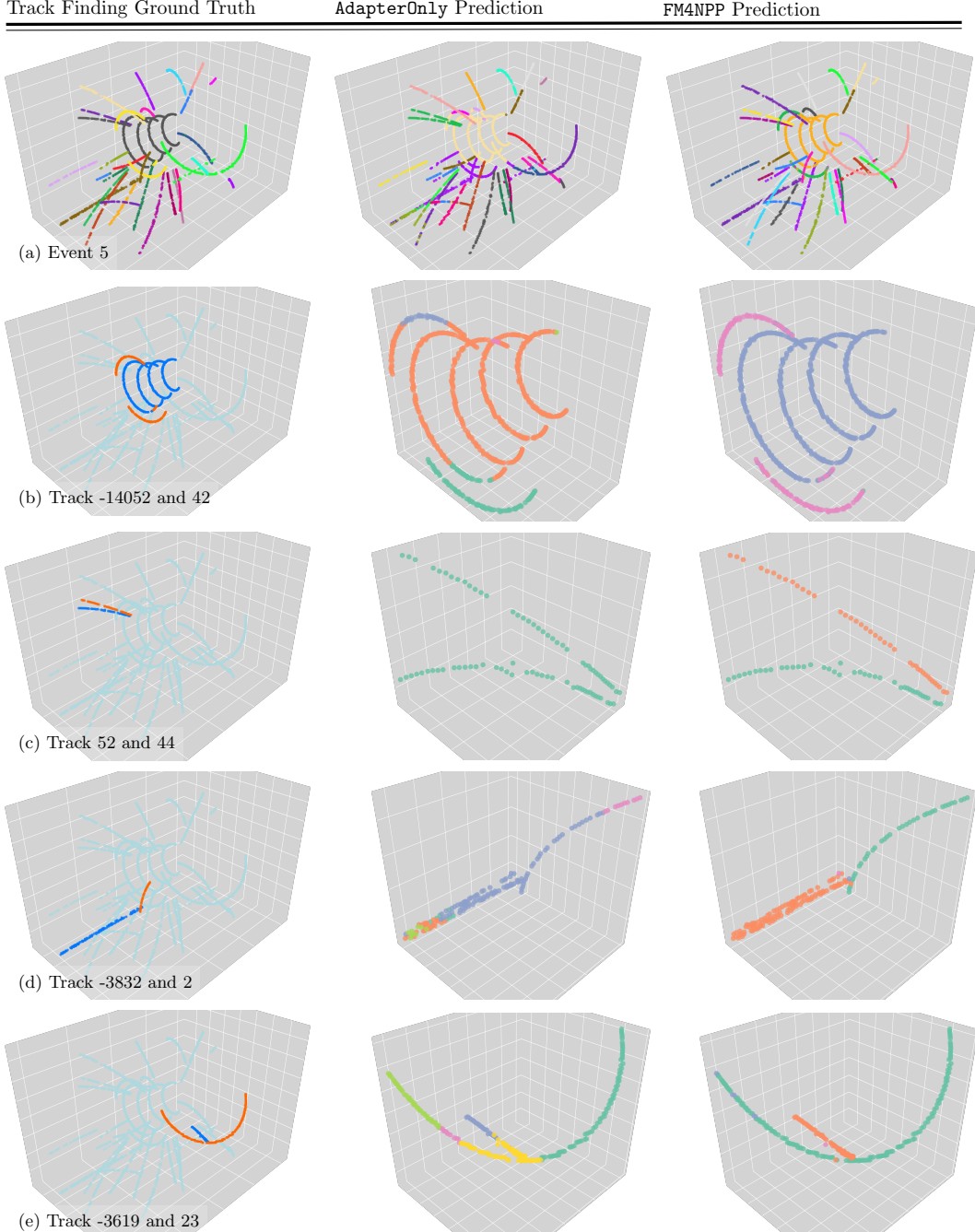

(a) Event 5

(b) Track -14052 and 42

(c) Track 52 and 44

(d) Track -3832 and 2

(e) Track -3619 and 23

Figure 12: **Performance of `AdapterOnly` and `FM4NPP` on track finding.** In panel (a), we show the ground-truth tracks, the `AdapterOnly` track candidates, and the `FM4NPP` track candidates (note that two different tracks might have the same color as the length of the color cycle used may be smaller than the number of tracks). In panel (b)-(e), we show four pairs of close-by ground-truth tracks that the `AdapterOnly` model fails to separate while the `FM4NPP` model does.

For each configuration, we train two variants: one with the pretrained backbone (frozen) and one without pretraining (replacing the backbone with a learnable linear layer) to isolate the effect of pretraining quality.

**Results.**    Figure 13 shows validation loss as a function of adapter capacity. Performance improves substantially when adding the first SA layer (0L → 1L: 45.7% loss reduction) with diminishing

returns thereafter. The optimal configuration uses two SA layers, achieving validation loss of 0.238. Adding more layers (4L) slightly degrades performance to 0.244, suggesting overfitting or that the adapter capacity exceeds what the frozen backbone can effectively support.

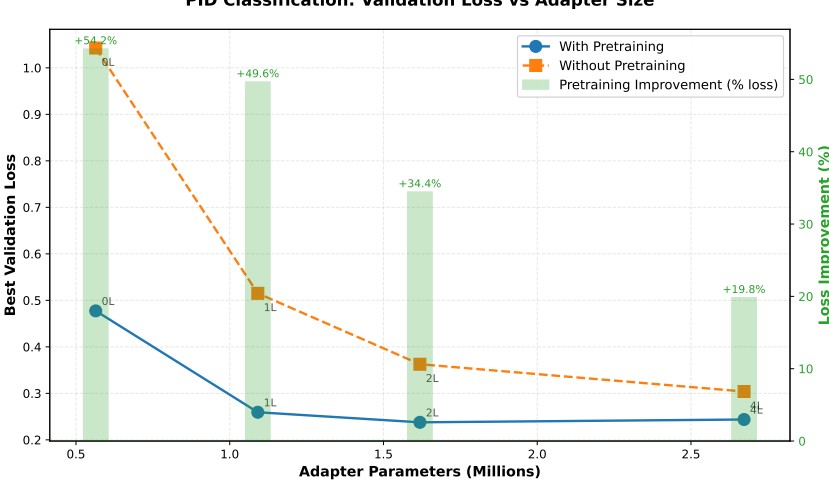

Figure 13: **Adapter capacity versus performance on PID task.** Validation loss decreases as adapter depth increases from zero to two self-attention layers then plateaus or slightly degrades at four layers. The gap between pretrained and non-pretrained models shrinks with larger adapters, indicating that small adapter heads are the performance bottleneck rather than foundation model quality.

**Interpretation.** The results reveal two key insights: 1) performance plateaus after two SA layers, indicating the frozen FM representation is already sufficiently rich – small adapter heads are the bottleneck rather than backbone quality. 2) While the relative pretraining benefit diminishes with larger adapters (54.2% for 0L decreasing to 19.8% for 4L), the pretrained backbone still provides substantial absolute improvements across all configurations. Even with the largest 4L adapter, the model with pretrained features achieves 0.244 validation loss compared to 0.304 without pretraining – a significant gap that underscores the effectiveness of the FM representation.

## C.5 ADDITIONAL LEARNED EMBEDDINGS RESULTS

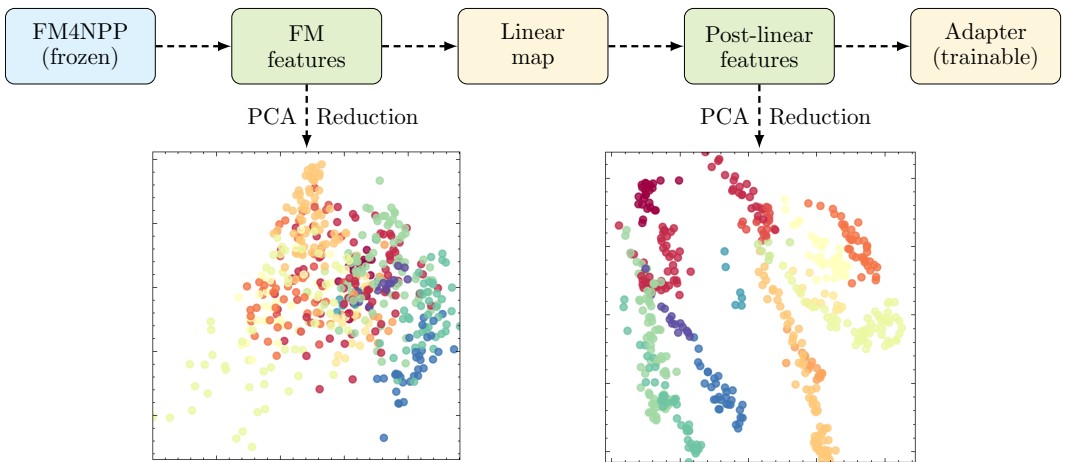

Figure 14: Visualization of learned embeddings from the foundation model (left) and the post-linear map features (right), projected via PCA reduction. Each marker corresponds to a spacepoint, colored by its associated track identity.

We analyze the neural embeddings from the frozen FM and their transformation after a simple linear projection, which precedes the lightweight adapter used for downstream tasks. To probe task specificity, we apply dimensionality reduction techniques (e.g., PCA) to both the raw FM embeddings and linearly projected features, focusing on a representative downstream task: track finding. As shown in Figure 14, the raw FM embeddings exhibit no clear separation among particle tracks, indicating the representations are task-agnostic. However, after applying a single linear projection, distinct and well-separated clusters emerge, corresponding to different particle tracks.

In Figure 15, we present results obtained by applying various dimensionality reduction techniques (including PCA, t-SNE, and UMAP) to both FM features and downstream adapter features. For illustrative clarity, we randomly select two test data samples. The results demonstrate consistent improvement, clearly showcasing the FM features' adaptability. Even after a single linear projection, the FM embeddings exhibit substantial clustering and separability, indicating rapid adaptation to the downstream track-finding task. Adapter features consistently provide superior discrimination, yielding distinctly well-separated clusters corresponding to different track categories. Because a linear transformation alone cannot create separability where none exists, this demonstrates the FM encodes rich, general-purpose information that only requires minimal alignment to become task-specific. It also explains why lightweight adapters, when built atop FM embeddings, outperform non-FM baselines by leveraging semantically meaningful input features.

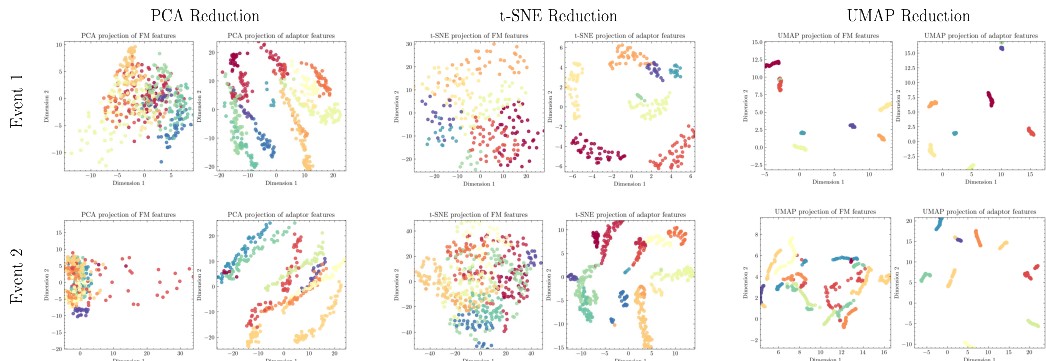

Figure 15: Dimensionality reduction results using PCA, t-SNE, and UMAP on randomly selected test data samples.

To further validate the robustness and generalizability of the FM features, we systematically investigate the impact of varying dimensionality reduction parameters using t-SNE. Specifically, we conduct experiments by setting the reduced dimensionality to three, four, and five and visualized the results by plotting the first two t-SNE components (See Figure 16). Across all tested dimensional configurations, the FM features consistently demonstrate pronounced clustering patterns and clear separability, highlighting their intrinsic adaptability and effectiveness in supporting diverse downstream classification tasks.

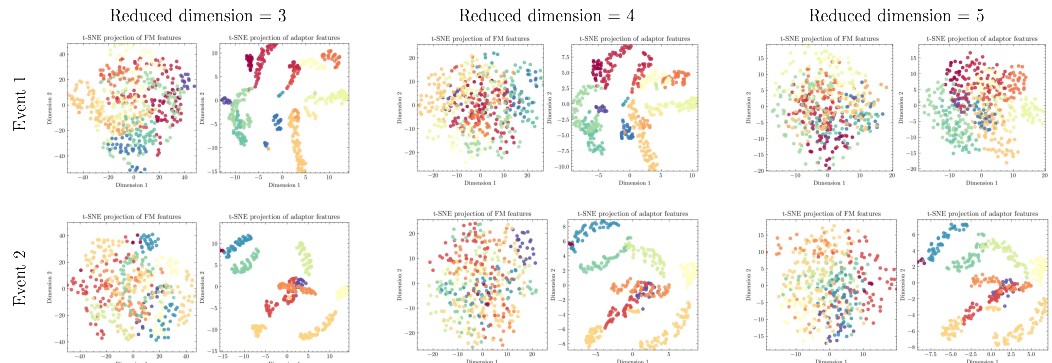

Figure 16: t-SNE visualizations for randomly selected test instances across various reduced dimensions.

In Figure 17, we extend our analysis to multiple downstream tasks, again using randomly selected test data instances and employing t-SNE for visualization. The FM features' separability was notably effective for the track-finding task, slightly diminished for particle identification, and considerably reduced for noise tagging. The limited performance observed in noise tagging is attributed to the inherent imbalance of the binary classification data, making separability challenging due to the dominant prevalence of a single label. Overall, our analyses confirm a hierarchy of effectiveness in FM embeddings across downstream tasks. Track-finding demonstrates the strongest separability, followed by particle identification, and lastly noise tagging. These findings align well with the FM's pretraining objective, neighbor identification, and are consistent with task relevance from a physics perspective.

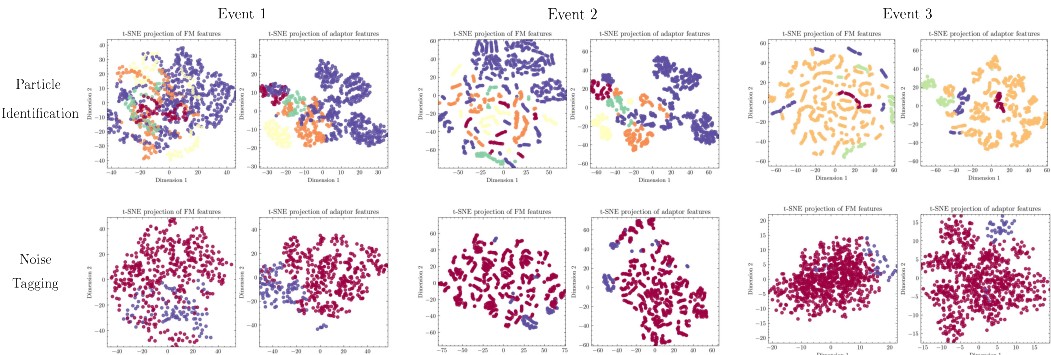

Figure 17: t-SNE visualizations for randomly selected test instances across various downstream tasks.

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
