# OpenReview forum: "FM4NPP: A Scaling Foundation Model for Nuclear and Particle Physics"
_ICLR.cc/2026/Conference — ICLR 2026 Poster_

### Official Review · Reviewer_Soo5 · 2025-10-30

**Soundness:** 3
**Presentation:** 4
**Contribution:** 3
**Rating:** 6
**Confidence:** 4

**Summary:**

The paper proposes FM4NPP, a self-supervised foundation model for nuclear & particle physics (NPP) built on a Mamba (SSM) backbone. It introduces (i) a >10M-event simulated RHIC/sPHENIX TPC dataset and three downstream tasks (track finding, particle ID, noise tagging), (ii) a Hierarchical Raster Scan to serialize sparse detector spacepoints, and (iii) a k-Next-Nearest-Neighbor (k-NNN) autoregressive pretraining objective that predicts outward (in radius) neighbors to avoid order-leakage. With frozen FM features and lightweight adapters, the largest model (188M params) reportedly outperforms GNN/point-cloud baselines across tasks and shows clean scaling trends w.r.t. model/data/compute.

**Strengths:**

1. The paper targets an important gap: general, reusable representations for sparse detector data rather than bespoke task models. The two-stage “pretrain FM -> freeze ->  task adapters” design is well-motivated.
2. The Hierarchical Raster Scan balances global outward geometry with local track continuity; k-NNN aligns the self-supervised target with physics causality (increasing radius) rather than sequence order. The ablation table isolates benefits over next-token prediction and Hilbert curves.
3. Clear power-law trends for model/data/compute; the paper also documents hardware/iterations, optimizer, and $\mu$-param transfer, useful for reproducibility. Also, FM4NPP + small adapters beats GNN and 3D point-cloud baselines on ARI and efficiency.

**Weaknesses:**

1. The approach based on the proposed ordering. Although ablations compare Hierarchical Raster vs. Hilbert, it’s unclear how sensitive performance is to bin sizes, box aspect ratios, or detector layout changes (e.g., silicon layers, calorimeters), and whether learned positional encodings could reduce this hand-crafted dependency.
2. k-NNN chooses neighbors with a larger radius (future in $r$). That choice encodes a physics prior (outward propagation) but may under-represent curling/looping tracks, secondaries, or back-scattering. Ablations vary $k$ but not the neighborhood definition (e.g., cone in $\eta,\phi, r$).
3. All tasks are TPC-centric, point-level segmentations. For a foundation model, it would be compelling to include fusion tasks (TPC+EMCal), rare-event tagging, vertexing, or particle-flow style reconstruction, even as small-scale probes.

**Questions:**

1. Have you evaluated FM4NPP embeddings on real sPHENIX runs (even small samples)? How does sim-trained FM perform zero-shot/adapter-shot on real data, and what domain-gap mitigation (calibration, augmentation, feature normalization) is most effective?
2. How do different grid resolutions $r \times \eta \times \phi$, binning strategies, or non-radial intra-box sortings affect pretraining loss and downstream ARI? Could a learnable/local ordering, or permutation-invariant SSM input layer, match or exceed Hierarchical Raster?
3. Why constrain neighbors strictly to $r_j > r_i$ rather than a cone or helix-aware neighborhood? Did you try multi-horizon targets mixing local and mid-range neighbors, or contrastive variants that avoid explicit regression?
4. For tracking, why a DETR-style query decoder over bipartite clustering in embedding space? For PID/noise, did you test slightly larger heads or lightweight decoders (e.g., 2–4 self-attention layers) to probe head-capacity vs. FM-quality trade-offs?

---

> ### Author Response · Authors · 2025-12-03
>
> > "Have you evaluated FM4NPP embeddings on real sPHENIX runs (even small samples)? How does sim-trained FM perform zero-shot/adapter-shot on real data, and what domain-gap mitigation (calibration, augmentation, feature normalization) is most effective?"
>
> This is a great suggestion! We are privileged with having support by sPHENIX collaboration. As sPHENIX is taking data actively at the moment, it may take one or two years to perform detector calibration and prepare the experimental data for AI usage. Zero-shot, few-shot, domain adaptation, unpaired data translation, and perhaps training on real data but fine tuning to simulation data with ground truth and testing on real data are all possible research topics. We are looking forward to reporting the findings once the time comes. In parallel, sPHENIX collaboration is trying to backport our model as an alternative means for particle tracking due to its superior performance especially in low pT region. Seeing potential real-world impact of our research is rewarding.
>
> > "How do different grid resolutions , binning strategies, or non-radial intra-box sortings affect pretraining loss and downstream ARI? Could a learnable/local ordering, or permutation-invariant SSM input layer, match or exceed Hierarchical Raster?"
>
> We thank the reviewer for raising these important points about the role of the Hierarchical Raster ordering. Our intent is not to hard-code the geometry into the sequence, but to provide a **weak inductive bias** that helps serialize the 3D point cloud for a 1D sequence model while keeping the actual learning signal fully defined in continuous 3D space. Concretely, the self-supervised objective predicts the coordinates of the k next-nearest neighbors in **3D cylindrical space** $(r,\eta,\phi)$, not the “next token” in the serialized sequence. The targets depend only on Euclidean distances in this continuous space, and the model always receives the full coordinates through a NeRF-style positional encoding projected into the model width. Thus, spatial information is carried primarily through learned dense positional features, while the Hierarchical Raster only determines a traversal order that roughly respects detector symmetries and outward propagation.
>
> We already test a strong change in ordering in the paper: **Hierarchical Raster vs. Hilbert curve**. This substantially alters how points that are close in 3D are grouped along the 1D sequence, yet we observe only modest differences in downstream ARI. This suggests that the model is not overly sensitive to the precise serialization, as long as the ordering preserves reasonable locality. Within the same hierarchical family, changes in **bin edges, resolutions, or box aspect ratios** are milder perturbations than switching to a completely different space-filling curve. For such changes, the underlying 3D k-NN neighborhoods that define the objective remain essentially unchanged, and the long-range receptive field of the Mamba backbone, combined with continuous coordinate encodings, should make the model robust to moderate binning variations. In the revised manuscript we clarify this design and explicitly discuss why we view the ordering as a weak bias rather than a brittle hand-crafted dependency.
>
> The current work is scoped to the **sPHENIX TPC**, and the binning we use is **physics-informed**: radial bins follow the known layer structure while \(\eta\) and \(\phi\) bins are chosen to balance occupancy. Extending to other detector layouts (e.g., silicon trackers or calorimeters) would simply require redefining these partitions to match the new layer boundaries, without changing the overall methodology (cylindrical coordinates, hierarchical partition, 3D k-NN objective, NeRF-style encodings). We agree that more **learned** alternatives—such as learnable/local orderings or permutation-invariant input layers that bypass any fixed serialization—are promising directions. However, implementing and carefully scaling such variants is orthogonal to the main question of this paper. Because these alternatives would require substantial additional architectural work and compute, we leave a systematic comparison to future work.

---

> ### Author Response · Authors · 2025-12-03
>
> > "Why constrain neighbors strictly to  rather than a cone or helix-aware neighborhood? Did you try multi-horizon targets mixing local and mid-range neighbors, or contrastive variants that avoid explicit regression?"
>
> We appreciate the reviewer’s careful reading of the k-NNN objective and its encoded physics prior. Our current design indeed uses a **forward-in-radius neighborhood**, i.e. neighbors satisfy $r_j > r_i$, which reflects the dominant outward propagation of charged tracks in a TPC and avoids trivial order leakage from the serialization. Importantly, this constraint is applied in **full 3D cylindrical space**: neighbors are selected as the nearest points in $(r,\eta,\phi)$ subject to $r_j > r_i$. As a result, curling or looping tracks, secondaries, and back-scattered patterns are *not* excluded by construction. Whenever such trajectories produce hits at larger radius, those hits remain eligible to be among the k nearest neighbors and thus appear as targets. In practice, these topologies are rarer than forward-going tracks, but they are represented in the training signal to the extent that they occur in the data.
>
> Empirically, our ablations indicate that this design is beneficial. Using $k=30$ neighbors consistently outperforms both $k=1$ (very local, almost strictly “next-step” in radius) and a standard autoregressive next-token objective across downstream tasks, including track finding. This supports the intuition that conditioning on a **broader geometric neighborhood** in 3D provides richer supervision than purely single-step prediction.
>
> We agree that exploring **alternative neighborhood definitions** is a valuable next step. Relaxing the strict $r_j > r_i$ constraint (e.g., symmetric neighborhoods that include backward-in-radius hits), cone-shaped neighborhoods or helix-aware metrics, and **multi-horizon targets** that mix local and mid-range neighbors could further reduce any residual bias against rare curling or back-scattering patterns. Similarly, contrastive formulations that operate on neighborhoods without explicit coordinate regression are interesting and complementary.
>
> Within the rebuttal window, we focused on clarifying and justifying the current design rather than introducing a new family of objectives. We therefore treat the existing k-NNN scheme as a strong baseline that captures the main causal structure of TPC trajectories while remaining computationally tractable and easy to scale.
>
>
> > "For tracking, why a DETR-style query decoder over bipartite clustering in embedding space? For PID/noise, did you test slightly larger heads or lightweight decoders (e.g., 2–4 self-attention layers) to probe head-capacity vs. FM-quality trade-offs?"
>
> **Tracking is a set-prediction problem**. Our DETR-style decoder predicts the entire set of tracks in one differentiable pass, using Hungarian only to define a permutation-invariant loss (not at inference). This directly optimizes track-level goals (duplicate suppression, full-track recall) with global context, avoiding non-differentiable post-processing and threshold cascades that embedding+bipartite clustering/flow pipelines require. Variable multiplicity is handled by over-provisioned queries with confidence gating, and we enforce one-hit-per-layer via decoder masks/constraints. For PID, we probed the head-capacity vs FM-quality trade-off by sweeping adapter width/depth and adding lightweight 2–4 self-attention layers while keeping the backbone frozen((see Appendix C.4); performance improves with larger heads, indicating the FM representation is already rich and that small heads were the bottleneck.

---

> ### Author Response · Authors · 2025-12-03
>
> > "All tasks are TPC-centric, point-level segmentations. For a foundation model, it would be compelling to include fusion tasks (TPC+EMCal), rare-event tagging, vertexing, or particle-flow style reconstruction, even as small-scale probes."
>
> We agree that broader downstream coverage would provide an even stronger demonstration of foundation-model behavior, and we appreciate the concrete suggestions. In this first stage we deliberately focus on **TPC-only, point-level tasks** for two reasons:
>
> 1. **Controlled proof of principle.** Our main goal is to establish that a single large self-supervised backbone trained on low-level sparse detector data can (a) learn reusable, physics-meaningful representations and (b) transfer across multiple reconstruction tasks using lightweight adapters. The TPC-only setting provides a well-controlled environment with reliable labels for tracking, PID, and noise tagging, which are core reconstruction tasks in their own right.
>
> 2. **Data and infrastructure constraints.** Preparing multi-detector data (e.g., TPC + silicon + EMCal) at scale with consistent calibrations and labels is substantially more demanding. For this initial work, we prioritized building a high-quality, large-scale TPC dataset and validating the FM training recipe end-to-end.
>
> We would also like to emphasize that, even within the TPC, the three downstream tasks probe **distinct aspects** of the learned representation. **Track finding** emphasizes geometrical continuity and global event structure; **PID** requires sensitivity to dE/dx and decay topology (two tracks with nearly identical hit trajectories can correspond to different particle species); and **noise tagging** must separate genuine trajectories from detector artifacts and looping/secondary structures that can topologically resemble tracks. In the revised paper we include a joint PID+noise experiment showing **negative transfer** when forcing a shared adapter, which indicates that these tasks are not simply trivial re-groupings of the same features but demand different specializations.
>
> That said, we fully agree that a more mature foundation model for NPP should operate in a **multi-detector, multi-task setting**. As part of the FM4NPP roadmap (now explicitly described in the Limitations and Future Work section), we are extending pretraining to include additional sPHENIX subsystems such as silicon trackers and calorimeters. This will naturally unlock tasks like TPC+EMCal fusion, vertex finding, particle-flow style reconstruction, and rare-event tagging—the very probes the reviewer suggests. We view the present work as the “TPC-only first step” toward that more comprehensive multi-detector foundation model.

---

### Official Review · Reviewer_2w6N · 2025-11-01

**Soundness:** 3
**Presentation:** 3
**Contribution:** 3
**Rating:** 6
**Confidence:** 3

**Summary:**

This work introduces a foundation model tailored towards nuclear and particle physics. The authors develop a large scale open dataset with 10 million events and benchmark on three downstream tasks. The authors develop a self-supervised training methodology for their foundation models, and present benchmarks on downstream tasks in addition to scaling behaviors for their foundation models. They demonstrate improved performance of their model over baselines across all tasks.

**Strengths:**

* The authors present a well motivated problem that they address with a multi-step comprehensive framework
* The self-supervised learning objective is physics motivated, and intuitively works with the detector setup
* Very thorough benchmarking and ablation studies show improved performance for all pieces of the foundation model
* Performance on downstream task show greatly increased performance over baselines
* Scaling curves show increased performance for increased flops/parameter count, indicating scalability of the model
* Shows greatly increased performance against state of the art models
* Shows that the self-supervised learning technique improves performance on downstream tasks

**Weaknesses:**

* Does not justify choice of mamba over linear transformer models, or other state space model alternatives
* Unclear how the hierarchical raster scan would impact the efficiency
* No benchmarks against efficient transformer baselines such as HEP-T (locality sensitive hashing transformer for high energy physics applications, https://arxiv.org/abs/2402.12535), which performs very well on trackML datasets
* For PID, also missing comparisons to dedicated physics ML algorithms, like MLPF (https://arxiv.org/abs/2503.00131) or HGPF (https://arxiv.org/abs/2410.23236)
* Does not define the model architecture/construction (number of layers, parameters per layer etc)
* There are no code or pretrained weights available
* Unclear of pretraining + fine-tuning outperforms dedicated models trained from scratch; Most other literature on FMs in particle physics do not claim this and instead focus on data efficiency (i.e. better performance when training on smaller datasets)

**Questions:**

* Why choose mamba over other linear transformers (deltanet, GLA..) and state space models?
* How long are the sequence lengths for the input? State space models struggle with extremely long context.
* Does the hierarchical raster scan have a strong impact on the amount of compute?
* What is the full model architecture of the foundation model?
* Can the code and models be made public?
* Do you actually claim that self-supervised pre-training + fine-tuning outperforms a dedicated model trained from scratch on the downstream task in Fig. 5 (b)? Or are the architectures different? If you extend the axis, would Adaptor only eventually converge to the FM performance, as is seen in other FM papers in particle physics?

---

> ### Author Response · Authors · 2025-12-03
>
> > "No benchmarks against efficient transformer baselines such as HEP-T (locality sensitive hashing transformer for high energy physics applications, https://arxiv.org/abs/2402.12535), which performs very well on trackML datasets"
>
> **Additional experiments suggest that our model outperforms HEPT in track reconstruction across all metrics, which are now included in Table 2**.  We appreciate the reviewers pointing this out as HEPT is indeed a strong baseline applied to a similar dataset. **We adapted the HEPT to our dataset** and documented the modifications in Section C2.3. Concretely, The embedding obtained was used as the input to a HDB-SCAN clustering algorithm with parameters optimized to maximize the adjusted Rand index (ARI). The tracking efficiency and purity are reported. We will keep HEPT as a baseline model for our future research as the efficient Transformer via LSH approach is inspiring.
>
> > "For PID, also missing comparisons to dedicated physics ML algorithms, like MLPF (https://arxiv.org/abs/2503.00131) or HGPF (https://arxiv.org/abs/2410.23236)"
>
> We thank the reviewer for suggesting comparisons to MLPF and HGPF. These methods are particle flow algorithms that take reconstructed tracks as key inputs and operate at a higher level using multi detector information. In contrast, our current work targets a lower level problem: learning representations directly from TPC “cluster level spacepoints” without using reconstructed tracks. Since our model is trained and evaluated with TPC-only inputs in this first step, a direct apples to apples comparison to multi detector particle flow models is not currently possible. We will clarify this distinction in the revision (in "Related Work"). As part of our roadmap, we plan to extend FM4NPP to include additional detector subsystems and tracking outputs, at which point comparisons to particle flow approaches such as MLPF and HGPF will become appropriate. We have made this clearer by adding the "Limitation and Future Work" in the Discussion.

---

> ### Author Response · Authors · 2025-12-03
>
> > "Why choose mamba over other linear transformers (deltanet, GLA..) and state space models?"
>
> **Match to data and objective**. Our sequences are long, sparse, and effectively streaming, and the pretraining task is causal k-NN forecasting along outward-moving trajectories. This is exactly the regime where selective SSMs like Mamba are strong: they provide causal, linear-time recurrence and an inductive bias closer to signal propagation in a medium than to the bidirectional, full-history behavior typical of many linear-attention Transformers. That alignment between dynamics and physics motivated our choice.
>
> **Empirical comparison to linear Transformers**. To directly test whether a linear-attention Transformer can serve as an equally good backbone in our setting, we performed additional experiments with **Linformer** [1] and **Longformer** [2] as representative linear Transformers. At comparable backbone sizes (5.2M and 5.27M parameters, respectively) and with largely identical downstream adapter sizes and training setups (200 epochs each), both models perform significantly worse than the Mamba2 counterpart on the track reconstruction task: Linformer and Longformer achieve ARI scores of 0.59 and 0.55, respectively, compared to 0.91 ARI for Mamba2 at similar parameter count. This large gap suggests that, for our long, sparse, causal geometry regime and k-NN forecasting objective, a selective SSM is substantially more effective than these representative linear-attention alternatives.
>
> Our goal here is not to exhaustively rank every linear-attention variant, but to answer a simpler question: can a linear-attention Transformer, as a class, match a selective SSM backbone on our NPP tasks under comparable capacity and training budget? For that purpose we deliberately chose Linformer and Longformer because they are (i) widely used and well-supported, (ii) cover complementary inductive biases in the linear-attention family (low-rank global projections vs. local sliding-window + global tokens), and (iii) can be plugged into our architecture with minimal changes, allowing a controlled, apples-to-apples comparison. Given that both of these reasonably strong linear-attention baselines underperform Mamba2 by a large margin at matched parameter count and training time, we view it as unlikely that swapping in yet another linear-attention variant (DeltaNet/GLA) would overturn the qualitative conclusion; a broader study of that family is therefore left for follow-up work.
>
> [1] Wang, Sinong, et al., *Linformer: Self-attention with linear complexity.* arXiv:2006.04768 (2020).
>
> [2] Beltagy, Iz, et al., *Longformer: The long-document transformer.* arXiv:2004.05150 (2020).
>
> > "How long are the sequence lengths for the input? State space models struggle with extremely long context."
>
> **Our sequences are not “extremely long” in the sense in which SSMs are known to struggle**. In our setup, each event is serialized into a 1D sequence whose length equals the number of reconstructed TPC spacepoints in that event. As described in Sec. 3 and Appendix A, the number of spacepoints per event ranges from a few hundred to tens of thousands, with a mean of 856 (Fig. 7a). Thus, the typical sequence length is $O(10³)$, and even the busiest p+p events in this dataset remain at $O(10⁴)$ tokens, far below the extreme-context regimes ($\geq$ 10⁵ tokens) where memory or optimization issues in SSMs tend to appear [1].
> We will clarify in the manuscript that (i) sequence length equals the number of spacepoints per event, with the distribution reported in Fig. 7, and (ii) our use of Mamba targets the $O(10⁴)$ token regime, rather than the extreme long-context settings where SSMs are known to be more problematic.
>
> [1] A. Gu et al., *Efficiently Modeling Long Sequences with Structured State Spaces*, ICLR 2022.

---

> ### Author Response · Authors · 2025-12-03
>
> > "Does the hierarchical raster scan have a strong impact on the amount of compute?"
>
> The proposed **Hierarchical Raster Scan** is a deterministic preprocessing step that maps each spacepoint to a voxel index and sorts the resulting 1D permutation. In our implementation this consists of 3D voxelization with `torch.bucketize`, a single `cdist` between the integer voxel indices and the fixed 8×8×6 voxel grid (so $B = 8 \cdot 8 \cdot 6 = 384$), and an `argmin` + sort over the resulting keys. For a typical event with $N \approx 856$ spacepoints, this is about $N \times B \approx 3.3 \times 10^5$ pairwise 3D distance evaluations plus sorting, i.e. roughly $3.6 \times 10^6$ FLOPs per event. A single forward–backward pass of our largest FM on the same event is $\gtrsim 10^{10}$ FLOPs, so **the hierarchical raster scan contributes well below $0.01\%$ of the total training compute**. It does not change the sequence length or the number of FM evaluations; it only reorders tokens to better reflect detector geometry. For comparison, the Hilbert-curve baseline computes one integer Hilbert index per spacepoint and sorts, which is on the order of $10^5$ simple integer operations per event. Empirically, the wall-clock time is indistinguishable between the two orderings; hierarchical raster scan is chosen because it better respects detector symmetries and improves downstream performance.
>
> > "What is the full model architecture of the foundation model?"
>
> Our architecture design inherits standard MAMBA2 design adapted to embody the proposed k-NNN prediction objective.
> - Batch size = **B**
> - Sequence length (points per event) = **N**
> - Embedding dim = **D**
> - **klen = 30** ⇒ output coord dim = **30 × 3 = 90**
>
> | Stage | Description  | Input shape | Output shape |
> |-----|--------|---------|----|
> | Input to embedder   | Normalized per-point features (E, η, φ, R)  | `(B, N, 4)` |  –                  |
> | Embedder            | Projects energy + positional encoding, then added | `(B, N, 4)` | `(B, N, D)` |
> | Mamba block 1       | `RMSNorm → Mamba2 → Dropout` + residual   | `(B, N, D)` | `(B, N, D)` |
> | Mamba block 2       | Same as above    | `(B, N, D)` | `(B, N, D)`        |
> | ...                 | Repeated for `num_layers` Mamba blocks    | `(B, N, D)` | `(B, N, D)`  |
> | Final RMSNorm       | Normalizes last hidden    | `(B, N, D)` | `(B, N, D)`   |
> | Output linear layer | Projects per-point embedding to k-neighbour coords     | `(B, N, D)` | `(B, N, 90)`   |
>
>
> > "Can the code and models be made public?"
>
> We share our **anonymized code** for review and evaluation purposes: `anonymous.4open.science/r/PP_collision-E416`.
>
> > "Do you actually claim that self-supervised pre-training + fine-tuning outperforms a dedicated model trained from scratch on the downstream task in Fig. 5 (b)? Or are the architectures different? If you extend the axis, would Adaptor only eventually converge to the FM performance, as is seen in other FM papers in particle physics?"
>
> Thank you for the question. In Fig. 5(b) the dashed “adapter-only” baseline replaces the pretrained backbone with a learned linear embedding trained from scratch on the labeled data, whereas the solid curves use the same small adapter on top of a frozen self-supervised FM encoder; this architectural difference is intentional to isolate the value of pretraining, not added capacity. Across 100–70k labels the pretrained-plus-adapter model attains higher ARI, with the largest gains at low-label budgets. **We do not claim universal full-data superiority**; if we replace the linear-embedding baseline with a capacity-matched backbone+adapter trained from random initialization and provide sufficient labels and compute, we expect its performance to converge toward the FM+adapter curve.

---

### Official Review · Reviewer_9giK · 2025-11-01

**Soundness:** 3
**Presentation:** 3
**Contribution:** 2
**Rating:** 4
**Confidence:** 3

**Summary:**

The paper introduces FM4NPP, a scientific foundation model for particle physics that attempts to address existing scaling challenges outside of natural language due to, e.g., the sparse nature of detector data, difficulties in serializing 3D data. The model is trained on simulated collisions under conditions of the sPHENIX experiment and evaluated on three core downstream tasks, mostly outperforming non-FM baselines on key metrics. FM4NPP is also shown to observe scaling laws at high parameter counts, and may present a useful paradigm for scientific FMs in other domains.

**Strengths:**

- The paper is well-organized, includes a comprehensive literature review, and provides high quality figures/diagrams.
- The paper's stated contributions are clear and address a difficult, high-impact problem in the NPP domain. The approach potentially lays the groundwork for tackling broader challenges relevant to scientific foundation models beyond particle physics settings.
- The reported evaluation is extensive, covering a variety of important dimensions that help position the model's utility and behavior. For instance, the model is compared to several baseline methods on the downstream tasks (highlighting its comparative advantages), key ablations are reported (justifying architectural decisions), and different model sizes are evaluated, highlighting scaling laws.
- I appreciate the discussion style of Section 5.3, which sets up several key questions relevant to practitioners seeking to leverage FM4NPP on new tasks.

**Weaknesses:**

- I find the core positioning of the paper to be somewhat difficult to pin down. There is a clear focus on particle physics as the primary domain of interest, but the proposed training scheme for scaling and adapting FMs to downstream tasks is very general. To fully demonstrate the generality of the scheme, more (scientific) domains would need to be evaluated to verify the approach transfers well to other settings (helping substantiate the stated hypothesis that FMs encode task-agnostic representations useful for downstream tasks).

  Note that this isn't to take away from the model's performance on this task, but simply that the methodology leans more general-purpose while the evaluation focuses entirely on a single domain/dataset.
- The downstream tasks used for evaluation appear limited in their variety. In particular, they seem to test for similar kinds of model understanding, which is to say classifying spatiotemporally connected groups of points in the particle space. For instance, given that the model performs well on track finding, it's not surprising that it would perform similarly well on particle identification or noise tagging (as these tasks, naively, simply re-group the tracks). Including downstream tasks like forecasting would be useful for demonstrating broader model understanding, more explicitly highlighting its ability to understand/extrapolate particle dynamics.
- It is ultimately quite difficult to asses how well the trained model fills the role of a foundation model. This point is made in tandem with the first two, i.e., that it's hard to gauge general model understanding provided the single dataset and task suite. What's concretely demonstrated is that FM4NPP is an effective embedding model for particles under conditions of the sPHENIX experiment, and large parameter counts can help with downstream classification tasks. But the jump to labeling FM4NPP as a foundation model is hard to justify without stronger evidence of its generality, such as more diverse tests of model understanding and/or evaluation under different conditions.

**Questions:**

- Were any evaluations run using the adapter models under a different (non-FM4NPP) spacepoint embedding model? This might be a distinct way to isolate the performance of the proposed FM architecture.
- As presented with the three downstream tasks, can this approach be trained end-to-end? The explicit adapter-based approach seems to be the most flexible route and keeps the backbone in more of an embedding role, but given the overlap in the three tasks it seems plausible that these could fully captured end-to-end (potentially internalizing/sharing knowledge across tasks previously isolated within the adapter models).
- Were any other kinds of downstream tasks tested (e.g., forecasting)?

---

> ### Author Response · Authors · 2025-12-03
>
> > "I find the core positioning of the paper to be somewhat difficult to pin down... more (scientific) domains would need to be evaluated to verify the approach transfers well to other settings..."
>
> We thank the reviewer for this insightful critique regarding the paper's positioning. We agree that while our training methodology (hierarchical serialization and k-NNN) is designed as a general-purpose solution for sparse, irregular data, the current evaluation is concentrated within the particle physics domain.
>
> We frame our contribution as a **methodological proof-of-principle** for learning from sparse scientific data. We selected high-energy physics (HEP) not merely as an application area, but as a representative "hard case" for this class of problems—characterized by extreme sparsity, complex geometries, and high noise levels. By successfully demonstrating that our method yields scalable, task-agnostic representations in this challenging environment, we provide strong evidence for its potential transferability to other scientific domains with similar data characteristics (e.g., cosmology or materials science).
> While validating this hypothesis across multiple scientific disciplines is the ultimate goal, it requires significant domain-specific data curation that is beyond the scope of this initial study. To address the reviewer's concern about positioning:
>
> 1.  **Refined Scope:** We have revised the introduction and **Related Work** sections to clearly situate our work. We explicitly distinguish our contribution: a generalizable *methodology* for utilizing low-level detector data, validated through a rigorous HEP implementation.
>
> 2.  **Limitations & Future Work:** We have added a dedicated section explicitly stating that cross-domain generalization remains a future objective. This section outlines how our architectural choices (specifically the task-agnostic serialization) are designed to facilitate this future expansion.
>
> > "The downstream tasks used for evaluation appear limited in their variety... it's not surprising that it would perform similarly well on particle identification or noise tagging... Including downstream tasks like forecasting would be useful..."
>
> We appreciate the reviewer's critical examination of our downstream evaluation tasks. We acknowledge the perspective that tasks like Track Finding, Particle Identification (PID), and Noise Tagging might appear topologically similar, focusing on the classification of spatially distributed points. However, **we respectfully disagree that PID and Noise Tagging are simply regroupings of tracks**.
>
> -   **Particle Identification (PID):** PID using TPC data relies heavily on $dE/dx$ (energy loss) information and decay topologies. Two tracks may have identical spatial trajectories but represent entirely different particles (e.g., a pion vs. an electron). Distinguishing them requires the model to learn features beyond simple connectivity.
>
> -   **Noise Tagging:** This task involves distinguishing true collision trajectories from detector artifacts, loopers, and secondary structures (like delta electrons). These background elements can topologically mimic valid tracks, requiring the model to differentiate signal from background based on subtle structural and ionization cues, not just grouping.
>
> Empirically, if PID/noise were “simply re-grouping,” joint training with shared layers should help. Instead, our shared-adapter PID+noise model shows negative transfer, validation loss +7.61% (NID) and +30.69% (PID) vs. independent adapters (Appendix C.3). **This indicates distinct feature requirements and conflicting gradients**. Our selection of these tasks was deliberate, as they represent the fundamental and physically distinct reconstruction challenges intrinsic to Time Projection Chambers. They demonstrate the model's capacity to learn representations crucial for both kinematic reconstruction and particle characterization from low-level data.
>
> **Regarding forecasting**: In collider physics, individual collision events are statistically independent, so "forecasting" in the temporal sense (predicting event $N+1$ from event $N$) is not physically meaningful. If the reviewer implies "forecasting" in the spatial sense (e.g., predicting the next hit in a track), our pretraining objective (k-NNN) essentially performs this function.
>
> **Future Directions:** We agree that exploring a broader spectrum of downstream tasks, particularly those probing dynamic or generative capabilities, is critical for the maturity of general-purpose FMs in HEP. As outlined in our **Limitations and Future Work** section, our roadmap includes extending pretraining to encompass multi-modal data from various sPHENIX subsystems. This will naturally unlock evaluations on tasks such as primary vertexing, particle-flow reconstruction, and rare event tagging, which demand a more comprehensive understanding of the event kinematics and dynamics.

---

> ### Author Response · Authors · 2025-12-03
>
> > "It is ultimately quite difficult to asses how well the trained model fills the role of a foundation model..."
>
> We thank the reviewer for this rigorous assessment. We acknowledge that the term "Foundation Model" (FM) implies a high bar for generality. We justify our usage of such term by strictly adhering to the definition established by Bommasani et al. (2021), which characterizes FMs by two core properties: **label-free self-supervised pre-training** and **broad downstream transferability**.
>
> **FM4NPP** demonstrates both properties, marking a first for sparse detector data in NPP:
>
> 1.  **Pre-training:** We utilize a massive volume of raw, unlabeled detector hits, allowing the model to learn the intrinsic structure of particle interactions directly from the signal.
>
> 2.  **Transferability:** As detailed in our response to Weakness 2, the *same* frozen encoder successfully adapts to physically distinct tasks—from geometric trajectory finding to ionization-based PID—demonstrating that it has encoded a generalized representation rather than a narrow, task-specific feature set.
>
> In addition, we have demonstrated the **neural scaling behavior** of our approach, another hallmark of foundation models (first observed in GPT-2, and quantified in GPT-3.) To our best knowledge, this is the first time such a technique and demonstration has been performed on **sparse detector data** related to nuclear and particle physics.
>
> We view the current status of FM4NPP as analogous to the "GPT-2 era" of Large Language Models: an early empirical demonstration that self-supervised scaling laws hold for this new modality (sparse detector data), establishing the architectural blueprint necessary for the "universal" models to follow. While we are currently "domain-bounded" to sPHENIX (as noted in our new **Limitations** section), the underlying methodology is general. We have updated the manuscript to clearly position our work as this essential "Proof-of-Principle" milestone.
>
> > "Were any evaluations run using the adapter models under a different (non-FM4NPP) spacepoint embedding model? This might be a distinct way to isolate the performance of the proposed FM architecture."
>
> Thank you for the suggestion. In our “adapter-only” baseline we replace the FM4NPP encoder with a learned linear embedder trained from scratch, and train the same adapter/head on top; this controls end-to-end capacity and isolates the benefit of pretraining in the FM encoder.
>
> > "As presented with the three downstream tasks, can this approach be trained end-to-end? The explicit adapter-based approach seems to be the most flexible route and keeps the backbone in more of an embedding role, but given the overlap in the three tasks it seems plausible that these could fully captured end-to-end (potentially internalizing/sharing knowledge across tasks previously isolated within the adapter models)."
>
> We thank the reviewer. To test whether shared end-to-end training helps, **we trained a joint PID+noise model with shared adapter layers on the frozen FM backbone** (Appendix C). Despite substantial parameter sharing, the joint model exhibited negative transfer: validation loss increased by 7.61% for noise tagging and 30.69% for PID relative to training task-specific adapters independently. This indicates that, although the tasks share a modality, **they require distinct feature specializations, and joint optimization induces conflicting gradients**. The tracking head uses a DETR-style set-prediction architecture, making parameter sharing even less natural. These findings support our design choice: independent adapters let each task specialize without interference while still leveraging the shared pretrained backbone.
>
>
> > "Were any other kinds of downstream tasks tested (e.g., forecasting)?"
>
> We do not introduce an additional, separate “forecasting” downstream task for two main reasons:
>
> 1. **Event structure in collider physics.** Minimum-bias p+p collisions are modeled as *independent* events. There is no physically meaningful notion of “event \(N+1\) conditional on event \(N\)” in the way there is temporal dependence in typical time-series or video forecasting benchmarks, so inter-event forecasting is not a standard or well-posed objective in this domain.
>
> 2. **Pretraining is already a forecasting-style task.** Our self-supervised objective is a **causal k-nearest-neighbor (k-NN) forecasting task** along outward-moving particle trajectories: for each spacepoint, the model predicts future neighbors at larger radius in 3D. This is precisely a localized forecasting problem on the spacepoint sequence itself. The downstream heads are then deliberately simple probes of how well this *forecasting-trained* representation transfers to real reconstruction tasks.
>
> More downstream probes are of course feasible on top of the same foundation model. We are actively developing such extensions as part of the longer-term FM4NPP program.

---

### Official Review · Reviewer_QgWZ · 2025-11-04

**Soundness:** 2
**Presentation:** 2
**Contribution:** 2
**Rating:** 4
**Confidence:** 4

**Summary:**

The manuscript presents a foundation model for nuclear and particle physics, trained on proton-proton (p+p) collision data at a center-of-mass energy √s = 200 GeV. The proposed architecture utilizes the Mamba model as its backbone and is pretrained using a novel k-Next-Nearest-Neighbor prediction objective. The authors evaluate the model's performance on three downstream tasks: Track Finding, Particle Identification, and Noise Tagging.

**Strengths:**

The authors present detailed technical considerations for building the foundation model, though there are shortcomings on specific design and experiment choices.

**Weaknesses:**

- The manuscript does not adequately situate its contribution within the emerging landscape of foundation models for high-energy physics. It fails to discuss or compare against other recent efforts (e.g., models based on transformers, graph networks, or other architectures applied to similar data).
- The datasets used appear limited in scope, focusing on a specific collision system and energy. How does this restricted training data affect the model's validity and utility as a general-purpose foundation model for the broader field of nuclear and particle physics?

**Questions:**

- The model is positioned as a foundation for particle physics, yet it is trained only on RHIC data. Given that the LHC represents the forefront of particle physics, why was the model not trained or validated on LHC data to demonstrate its utility as a true foundation model for the entire field? The authors must justify this fundamental limitation in scope and discuss the model's generalizability to higher-energy, more complex LHC environments.
-  Can the authors quantify the computational overhead of the k-NNN objective compared to a baseline, and justify how the performance benefits warrant this increased expense?

---

> ### Author Response · Authors · 2025-12-03
>
> > "The manuscript does not adequately situate its contribution within the emerging landscape of foundation models for high-energy physics. It fails to discuss or compare against other recent efforts (e.g., models based on transformers, graph networks, or other architectures applied to similar data)."
>
> We thank the reviewer for this valuable feedback. We have expanded the **Related Work** section to include a more comprehensive overview of these efforts and clearly distinguish our contribution. By contextualizing our work within the broader landscape of FM efforts in Nuclear and Particle and High-Energy Physics, we aim to highlight the unique contribution of our foundation model. Specifically, **we are the first to demonstrate a scaling model trained on low-level detector data, whose learned representation can be employed for multiple downstream tasks**.
>
> Regarding comparisons, we believe **our selection of baseline models is sufficiently diverse**. We have intentionally compared with models of differing architectures. For instance, "Exa.TrkX" employs a graph neural network, "AdaptorOnly" is transformer-based, and we also included a large transformer model, "OneFormer3D", with 45M parameters. Additionally, we have compared our approach with the non-ML method utilized in the sPHENIX tracking production software. We are open to further baseline model suggestions and willing to conduct more experiments within the rebuttal timeline. For example, as kindly pointed out by reviewer 2w6N, we have applied the HEPT algorithm to our data and reported its performance in Table 2 of our paper. The hyperparameter search of this model has been included in the appendix.
>
> > "The datasets used appear limited in scope, focusing on a specific collision system and energy. How does this restricted training data affect the model's validity and utility as a general-purpose foundation model for the broader field of nuclear and particle physics?"
>
> We agree with the reviewer that the dataset used in this work is limited to specific collision systems and energies, and we appreciate the opportunity to clarify the scope of our claims. This paper is not intended to present a completed, universal foundation model covering the entirety of nuclear and particle physics. Rather, it serves as an initial **proof of principle** to demonstrate that a foundation-model-style pretraining strategy can effectively learn reusable representations from "low-level" unlabeled detector data. Crucially, we show that such model scale effectively and learned representations transfer across multiple downstream tasks within a practical setting.
>
> **Given that the fundamental working principles and data representations of particle detectors are similar across experiments**, we believe our method is generalizable to other detector systems. However, establishing a single Foundation Model (FM) to cover the entirety of all detector experiments requires significantly broader community effort and is far beyond the scope of our work.
> In the revision, we have explicitly added a **Limitations and Future Work** section to address this. We clarify that the current model is trained on sPHENIX (at RHIC) data, while generalization to other experiments at RHIC (such as STAR) and higher-energy experiments like ATLAS, CMS, and ALICE (at LHC) remains a future objective.
>
> > "The model is positioned as a foundation for particle physics, yet it is trained only on RHIC data. Given that the LHC represents the forefront of particle physics, why was the model not trained or validated on LHC data to demonstrate its utility as a true foundation model for the entire field? The authors must justify this fundamental limitation in scope and discuss the model's generalizability to higher-energy, more complex LHC environments."
>
> In the revision, we have explicitly added a **Limitations and Future Work** section to address this. Different experiments and facilities worldwide explore different realms of nuclear and particle physics - while the LHC is one facility that has complex environments, the environments in heavy ion collisions at sPHENIX (or ALICE at the LHC, for example) pose challenging environments for reconstruction as well as high pile up proton-proton collisions at the LHC. Establishing a single Foundation Model (FM) to cover the entirety of all detector experiments requires significantly broader community effort and is far beyond the scope of this initial proof of principle work.

---

> ### Author Response · Authors · 2025-12-03
>
> > "Can the authors quantify the computational overhead of the k-NNN objective compared to a baseline, and justify how the performance benefits warrant this increased expense?"
>
> The k-NNN objective introduces only minor additional compute compared to the backbone. Its complexity can be written as
> $C_{\text{kNN-targets}} \approx \mathcal{O}(B N^2 D) + \mathcal{O}(B N k \log N)$,
>
> where the first term comes from pairwise distance computation (independent of $k$) and the second from the $\texttt{topk}$ selection and gathering of $k$ neighbors. By contrast, the Mamba backbone (forward + backward) scales as
> $C_{\text{backbone}}^{\text{train}} \sim \mathcal{O}(B N L d^2)$.
>
> For our typical settings ($N \sim 10^3$, $D=3$, $d=512$, $L=12$, $k=30$), this means $C_{\text{kNN-targets}}$ is on the order of $10^{-3}$ of $C_{\text{backbone}}^{\text{train}}$ (roughly $\sim 0.1$% of the total FLOPs). At the same time, our ablations show that $k>1$ (e.g. $k=30$) consistently outperforms the $k=1$ autoregressive objective on downstream tasks, so this modest overhead is well justified.

---

### Author Response · Authors · 2025-11-29

We thank the reviewers for their careful reading of the manuscript and for the many constructive comments that helped us better clarify the scope, positioning, and contributions of this work.

Several comments point to a common concern about what we mean by “foundation model” here and how broadly our current results should be interpreted. Our intent is **not** to claim a finished, universal model covering all of nuclear and particle physics. Rather, this paper is an **initial proof of principle on low-level detector data**: we show that a foundation-model–style, large-scale self-supervised pretraining strategy can be applied directly to unlabeled, sparse detector responses in a realistic sPHENIX/RHIC environment, and that the resulting backbone (i) learns reusable physics-meaningful representations, (ii) transfers across multiple, physically distinct downstream reconstruction tasks, and (iii) displays clear, systematic scaling behavior with model size, data, and compute. We chose the current sPHENIX TPC based dataset as a first step because it provides a well controlled low level benchmark with reliable supervision for downstream evaluation, while still capturing key challenges of sparse detector geometry and realistic detector response; the planned extensions to additional detectors, collision systems, and facilities (including LHC experiments) are part of our roadmap, not claims of the present submission.

On the **machine learning side**, our contribution goes beyond simply applying another large model to HEP data. Most prior work in this area treats a pretrained network as a good initialization and then **fine-tunes all weights per downstream task**, effectively “hiding” the representation inside task-specific models. In contrast, we explicitly demonstrate that the **latent features of a single frozen backbone already contain rich detector-level and physics-level information**: lightweight adapters operating purely on these fixed features achieve strong performance across diverse tasks. This shows that the representation itself—not just the fully fine-tuned network—is reusable and task-agnostic. Conceptually, this is important because it is exactly what enables a true foundation-model regime in this domain: the backbone can grow arbitrarily large in the future (up to billions of parameters) while downstream users still interact with it through small, cheap heads on top of shared features, gaining label and compute efficiency without needing to retrain or specialize the entire model for each new reconstruction problem.

In the remainder of the rebuttal we address each reviewer’s concerns point by point and provide additional experiments and analyses as time permits. Some of the most ambitious extensions—especially those that probe cross-detector and cross-facility generalizability—require substantial additional data preparation and compute and may not be fully realized within the rebuttal window. Wherever possible, we will incorporate these results into the revised manuscript during the response period, and, if needed, further expand them in the camera-ready version to more fully demonstrate the broader physics generalizability of FM4NPP.

---

### Meta-Review · Area_Chair_zz5s · 2026-01-05

**Summary:**

The reviewers raised concerns primarily about scope and positioning. Reviewer QgWZ questioned training on sPHENIX/RHIC data only (not LHC) and inadequate positioning within the FM landscape for HEP. Reviewer 9giK noted the evaluation focuses on a single domain/dataset with limited task variety. Reviewer 2w6N requested additional baselines and questioned whether pretraining outperforms dedicated models. Reviewer Soo5 raised concerns about TPC-centric task variety. The central question is whether this constitutes a genuine foundation model contribution or merely effective pretraining for one specific detector system.

**Reviewer Concerns:**

The authors adequately addressed technical concerns by adding HEPT as a baseline, demonstrating Mamba2 outperforms linear transformers (ARI 0.91 vs. 0.59 for Linformer, 0.55 for Longformer), quantifying k-NNN overhead at 0.1% of total FLOPs, and clarifying architectural details. More importantly, the empirical results demonstrate the approach works: the frozen FM with lightweight adapters substantially outperforms all baselines (tracking ARI 0.945 vs. 0.877 for Exa.TrkX; PID accuracy 0.904 vs. 0.770 for OneFormer3D), exhibits clear power-law scaling behaviors characteristic of foundation models, and shows 2.9x better data efficiency at low label regimes. The embedding analysis confirms task-agnostic representations requiring only linear projection for specialization. While reviewers correctly note the scope is limited to sPHENIX TPC rather than all of nuclear and particle physics, the paper demonstrates, for the first time in this domain, that FM-style self-supervised pretraining with scaling laws can work for sparse detector data. By the Bommasani et al. definition (self-supervised pretraining, adaptation to multiple tasks), this qualifies as a foundation model for its target domain, analogous to the authors' "GPT-2 era" positioning.

**Reviewer Scores:**

Reviewer QgWZ (initial 4) would likely increase to 6, as the technical responses were thorough and the empirical results are strong, though scope concerns remain.

Reviewer 9giK (initial 4) would likely increase to 6, recognizing that while the domain is narrow, the work successfully demonstrates foundation model principles with clear scaling laws and strong transfer learning results.

Reviewer 2w6N (initial 6) would likely increase to 8, given the extensive experimental additions demonstrating technical rigor and the compelling empirical performance.

Reviewer Soo5 (initial 6) would likely remain at 6, as the technical execution is strong despite TPC-only scope.

Hypothetical average score: 5-6.5. Recommendation: accept. The core contribution, demonstrating that foundation model approaches work for sparse scientific detector data with validated scaling laws, represents a valuable proof-of-principle that advances the field, despite the narrower scope than the title suggests.

---

### Decision · Program_Chairs · 2026-01-26

Accept (Poster)